# Identification of regulators of poly-ADP-ribose polymerase inhibitor response through complementary CRISPR knockout and activation screens

Kristen E. Clements[1], Emily M. Schleicher[1], Tanay Thakar [1], Anastasia Hale [1], Ashna Dhoonmoon[1], Nathanial J. Tolman[1], Anchal Sharma[2], Xinwen Liang[3], Yuka Imamura Kawasawa[1,4,5], Claudia M. Nicolae [1], Hong-Gang Wang[3,4], Subhajyoti De [2] & George-Lucian Moldovan [1✉]

Inhibitors of poly-ADP-ribose polymerase 1 (PARPi) are highly effective in killing cells deficient in homologous recombination (HR); thus, PARPi have been clinically utilized to successfully treat BRCA2-mutant tumors. However, positive response to PARPi is not universal, even among patients with HR-deficiency. Here, we present the results of genome-wide CRISPR knockout and activation screens which reveal genetic determinants of PARPi response in wildtype or BRCA2-knockout cells. Strikingly, we report that depletion of the ubiquitin ligase HUWE1, or the histone acetyltransferase KAT5, top hits from our screens, robustly reverses the PARPi sensitivity caused by BRCA2-deficiency. We identify distinct mechanisms of resistance, in which HUWE1 loss increases RAD51 levels to partially restore HR, whereas KAT5 depletion rewires double strand break repair by promoting 53BP1 binding to double-strand breaks. Our work provides a comprehensive set of putative biomarkers that advance understanding of PARPi response, and identifies novel pathways of PARPi resistance in BRCA2-deficient cells.

[1] Department of Biochemistry and Molecular Biology, The Pennsylvania State University College of Medicine, Hershey, PA 17033, USA. [2] Rutgers Cancer Institute of New Jersey, Rutgers the State University of New Jersey, New Brunswick, NJ 08901, USA. [3] Department of Pediatrics, The Pennsylvania State University College of Medicine, Hershey, PA 17033, USA. [4] Department of Pharmacology, The Pennsylvania State University College of Medicine, Hershey, PA 17033, USA. [5] Institute for Personalized Medicine, The Pennsylvania State University College of Medicine, Hershey, PA 17033, USA. ✉email: glm29@psu.edu

dentifying vulnerabilities specific to cancer cells remains the central goal of much research in the field of cancer biology; when successful, this line of research can significantly improve patient survival. In 2005, two groups discovered that the very mutations that contribute to tumorigenesis in some patients—mutations in the homologous recombination (HR) proteins BRCA1 and BRCA2—also sensitized cells to treatment with inhibitors of poly-ADP-ribose polymerase 1 (PARPi)[1,2]. The classical model explaining this synergy proposes that these drugs induce double-strand DNA breaks (DSBs) which cannot be repaired in BRCA-deficient cells[1,2]. In the following years, PARPi has shown remarkable promise; indeed, the FDA has approved PARPi for use in the clinic based on multiple clinical trials demonstrating an impressive improvement in progression-free survival (PFS)[3,4]. For example, in one study, olaparib treatment in ovarian cancer patients harboring a germline BRCA1/2 mutation resulted in 19 months PFS compared to the 5.5 months of those receiving placebo[3]. Given the impressive success of PARPi in this population of patients, there is great interest in identifying cohorts of BRCA1/2-proficient patients who may be able to benefit from this therapy, as well[5].

However, even though a large number of patients experience great clinical benefit from PARPi treatment, a significant sub-population either do not respond or quickly develop resistance to PARPi, despite being predicted to be sensitive based on their BRCA2 status[3]. In light of this, efforts have been made to understand the roles of individual proteins in mediating resistance to PARPi in the background of BRCA2 deficiency. For example, depletion of RADX or EZH2 has been shown to rescue the cytotoxicity caused by PARP inhibition in BRCA2- deficient cells[6,7]. Moreover, depletion of PTIP or CHD4 reduces PARPi-induced chromosomal aberrations in BRCA2-deficient cells[8,9]. However, a comprehensive, genome-wide characterization of potential mediators of PARPi sensitivity and resistance would both advance the fundamental understanding of the processes underlying these effects, as well as potentially promote more effective usage in the clinic.

In order to better understand the mechanisms regulating cellular sensitivity and resistance to PARP inhibitors, we designed complementary genome-wide CRISPR screens in a pair of parental wildtype and BRCA2-knockout (BRCA2$^{KO}$) HeLa cell lines. This approach allowed us to investigate which specific genetic changes lead to PARPi sensitivity in inherently resistant cells (parental) or to resistance in intrinsically sensitive cells (BRCA2$^{KO}$) in an otherwise isogenic background. We validate two proteins previously unconnected to PARPi resistance, namely KAT5 and HUWE1, which cause resistance to PARPi when depleted in BRCA2-deficient cells. Furthermore, we show that the mechanism of resistance caused by KAT5 depletion involves 53BP1-mediated regulation of end resection at DSBs. In contrast, loss of HUWE1 restores HR in BRCA2-deficient cells by increasing RAD51 levels.

## Results

### CRISPR screens to identify determinants of PARPi response.
In order to gain a broader understanding of factors governing PARPi response, we performed a series of genome-wide CRISPR knockout and transcriptional activation (overexpression) screens in an isogenic pair of wildtype and BRCA2-knockout HeLa cell lines. Employing these isogenic lines allowed for a more complete assessment of PARPi response, as the elimination of other potentially confounding mutations found in patient-derived cell lines facilitated the direct comparison of results among screens. Two independent replicates were performed for each screen.

First, to identify genetic changes which sensitize cells to PARPi treatment, we employed the Brunello human CRISPR knockout pooled library, which targets 19,114 genes with four single-guide RNAs (sgRNAs) per gene[10]. HeLa cells infected with the Brunello library were divided into PARPi (5 μM olaparib)- or vehicle (DMSO)- treated arms, and after 4 days surviving cells were harvested for sgRNA sequencing and bioinformatic analysis using the MAGeCK (Model-based Analysis of Genome-wide CRISPR/Cas9 Knockout) algorithm[11] (Fig. 1a). By treating parental HeLa cells with a relatively low dose of olaparib and identifying sgRNA sequences which dropped out in the olaparib-treated arm as compared to the DMSO-treated arm, we were able to generate a list of genes which, when knocked out, sensitize wildtype cells to PARPi (Fig. 1b, Supplementary Data 1). Among the top hits identified in this screen were RAD51, an essential component of the HR pathway, and its paralogs RAD51B, RAD51C, RAD51D and XRCC3. Indeed, RAD51C loss via mutation or promoter hypermethylation in tumors has been connected to favorable PARPi response in patients[12]. Other notable top hits include multiple RNase H2 subunits, consistent with previous findings implicating loss of RNase H2 in sensitizing HR-proficient cells to PARPi[13]. Collectively, top hits were enriched for biological processes including HR, DNA replication, and DNA repair (Supplementary Fig. 1).

Next, we sought to identify genes which cause PARPi resistance when depleted in BRCA2$^{KO}$ cells. To identify resistant cells, we treated CRISPR knockout library-infected BRCA2$^{KO}$ cells with a dose of olaparib (4 μM) which killed at least 90% of cells over 4 days (Fig. 1c). Then, we searched for sgRNA sequences which were enriched in the cells surviving olaparib treatment as compared to DMSO (Fig. 1d, Supplementary Data 2). A top hit from our screen was the transcription factor E2F7. We previously showed that loss of E2F7 causes PARPi resistance in BRCA2-deficient cells through transcriptionally upregulating RAD51 and subsequently restoring HR[14]. Moreover, it has been previously established that PARP1 itself is required for the cytotoxicity of PARPi[15–17]. In line with this, we found that PARP1 was ranked highly in our screen. Similar to the sensitivity screen, pathway analyses indicated that DNA replication and DNA repair processes were enriched among the top hits from the resistance screen (Supplementary Fig. 2). However, unlike as seen in the sensitivity screen, the pathway analysis of the resistance screen also implicated more unexpected pathways such as RNA processing and protein translation. While this finding could reflect a role for these pathways in PARPi resistance, an alternative explanation may be that proteins in these pathways have heretofore underappreciated roles in DNA repair.

Finally, we performed a CRISPR activation screen to identify genes which cause PARPi resistance when overexpressed in BRCA2$^{KO}$ cells. We utilized the Calabrese human CRISPR activation library, which transcriptionally activates 18,885 genes individually by using sgRNA sequences to recruit an enzymatically dead Cas9 (dCas9) and transcriptional activators to the region near the transcriptional start site of the gene[18]. Screening conditions were maintained as performed in the resistance CRISPR knockout screen. In brief, HeLa BRCA2$^{KO}$ cells expressing dCas9 were infected with the activation library and divided into high dose PARPi (4 μM olaparib)- or vehicle (DMSO)- treated arms (Fig. 1e, Supplementary Fig. 3a). Roughly 85% of cells were killed after 4 days of treatment; surviving cells were harvested and analyzed to identify genes which cause resistance to PARPi when transcriptionally activated (Fig. 1f, Supplementary Data 3). We further performed pathway analysis using the results of this screen alone, or pooled with the results of the similarly designed CRISPR knockout screen described above (Supplementary Fig. 3b, c, Supplementary Fig. 4). There are few

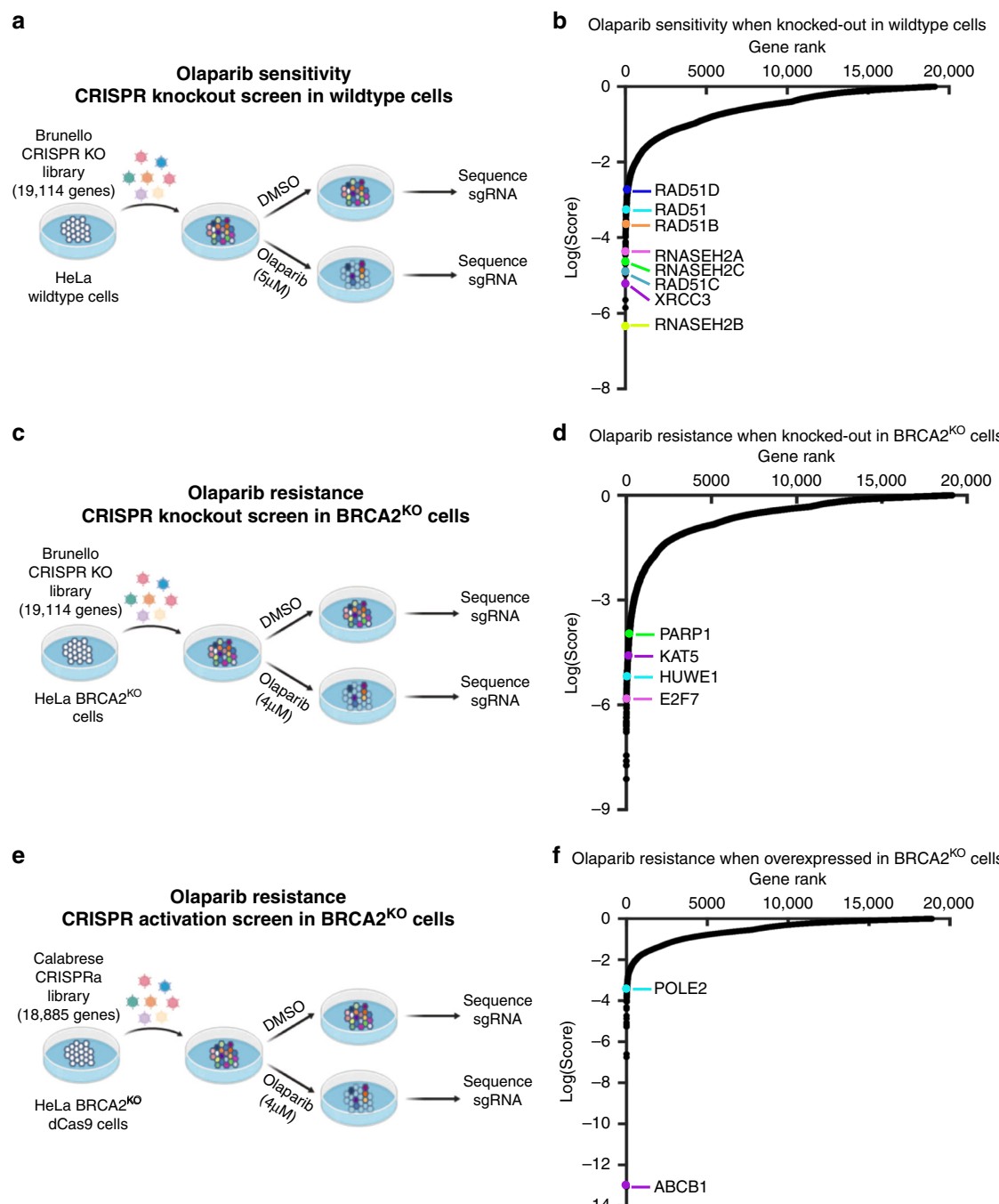

**Fig. 1 Complementary CRISPR knockout and activation screens identify determinants of PARPi response in parental or BRCA2-knockout HeLa cells.**
**a** Schematic representation of the CRISPR knockout screen for olaparib sensitivity in wildtype cells. HeLa cells were infected with the Brunello CRISPR knockout library. Infected cells were divided into PARP inhibitor (olaparib)-treated or control (DMSO) arms. Genomic DNA was extracted from cells surviving the drug treatment and single-guide RNAs (sgRNAs) were identified using Illumina sequencing. **b** Scatterplot showing the results of this screen. Each gene targeted by the library was ranked based on the MAGeCK negative selection score. Several biologically interesting hits are highlighted. **c** Schematic representation of the CRISPR knockout screen for olaparib resistance in BRCA2$^{KO}$ cells. HeLa BRCA2$^{KO}$ cells were infected with the Brunello CRISPR knockout library. Infected cells were divided into PARP inhibitor (olaparib)-treated or control (DMSO) arms. **d** Scatterplot showing the results of this screen, with several biologically interesting hits highlighted. Each gene targeted by the library was ranked based on the MAGeCK positive selection score. **e** Schematic representation of the CRISPR activation screen for olaparib resistance in BRCA2$^{KO}$ cells. HeLa BRCA2$^{KO}$ cells stably expressing the modified dCas9 enzyme were infected with the Calabrese CRISPR activation library. Infected cells were divided into PARP inhibitor (olaparib)-treated or control (DMSO) arms. **f** Scatterplot showing the results of this screen, with several biologically interesting hits highlighted. Each gene targeted by the library was ranked based on the MAGeCK positive selection score. Source data are provided as a Source Data file.

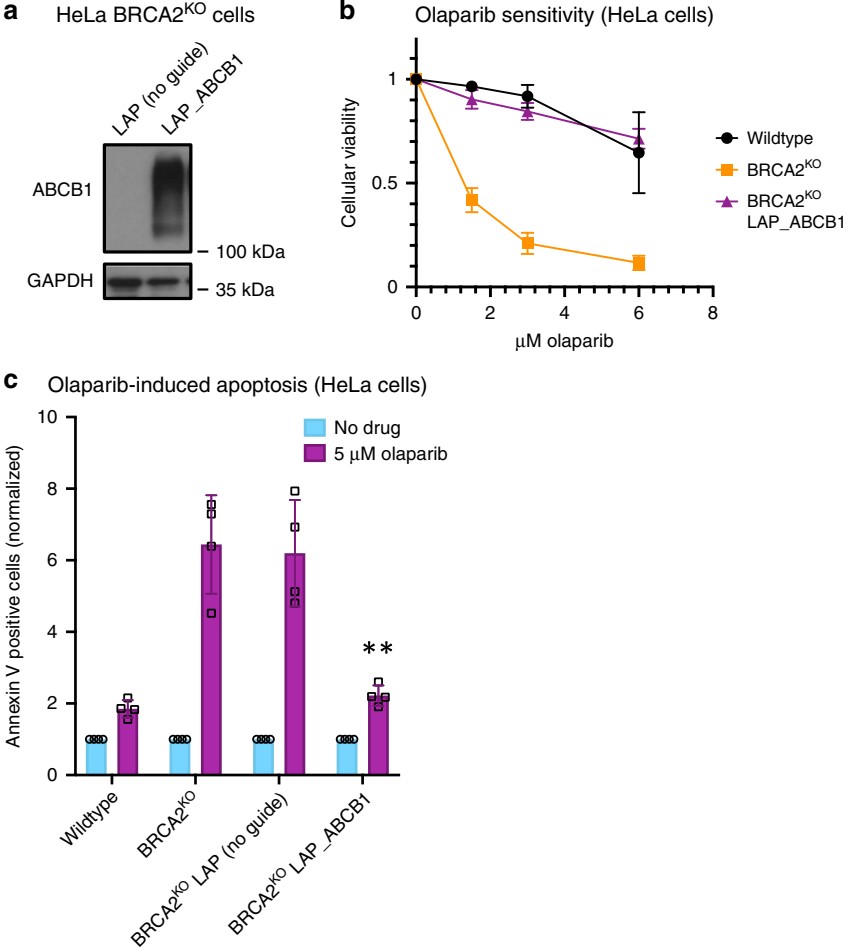

**Fig. 2 Overexpression of ABCB1, the top hit from the CRISPR activation screen, causes PARPi resistance in BRCA2-deficient cells. a** Western blot showing overexpression of ABCB1 in the cell line containing all three components of the CRISPR lentiviral activation particle (LAP) system (dCas9, activator helper complex, and sgRNA targeting the ABCB1 gene) but not in the control cell line lacking the sgRNA. **b** Cellular viability assay showing that ABCB1 transcriptional activation rescues PARPi sensitivity in HeLa BRCA2-knockout cells. The averages of four experiments are shown, with standard deviations as error bars. **c** Olaparib-induced apoptosis in BRCA2-knockout cells is suppressed by ABCB1 overexpression. The averages of four experiments are shown, with standard deviations as error bars. Asterisks indicate statistical significance (*t*-test, two-tailed, unequal variance) compared to the No Guide sample. Source data are provided as a Source Data file.

studies reporting proteins which cause PARPi resistance when overexpressed; however, interestingly, the top hit from our screen, ABCB1, has previously been identified as a mechanism of PARPi resistance[19]. This gene encodes the protein MDR-1 (multidrug resistance protein 1), a drug efflux pump whose overexpression was associated with acquired resistance to olaparib in ovarian cancer cell lines[19]. To validate these results in our system, we generated HeLa BRCA2KO cells with transcriptional activation of the ABCB1 gene (Fig. 2a). We found that the BRCA2-knockout, ABCB1-overexpressing cell line was as resistant to PARPi as the BRCA2-proficient parental HeLa line (Fig. 2b). Previously, PARPi treatment has been shown to induce apoptosis in BRCA2-deficient cells[20]. In line with this, we found that olaparib treatment led to a more than sixfold increase in cells positive for annexin V, a marker of apoptosis, in BRCA2-knockout cells; however, overexpression of ABCB1 restored olaparib-induced apoptosis to control levels (Fig. 2c). Overall, these findings validate the results of our CRISPR activation screen.

**Depletion of KAT5 or HUWE1 confers PARPi resistance.** Analysis of the most highly significantly enriched genes from the PARPi resistance knockout screen in BRCA2KO cells revealed

that 23 genes of the 214 top hits with false discovery rate (FDR) lower than 5% are associated with DNA replication and repair processes (Supplementary Fig. 5). Out of these, two DNA repair genes previously unconnected to PARPi resistance, namely the histone acetyltransferase KAT5 (also known as TIP60) and E3 ubiquitin ligase HUWE1, drew our attention. Both proteins have roles in cellular processes related to DNA replication and repair and are potential interactors of PARP1[21,22], yet they have not been directly connected to PARPi sensitivity. We first sought to validate these two hits. In order to test the effect of KAT5 or HUWE1 loss on PARPi sensitivity in BRCA2-depleted cells, we knocked down these genes and assessed cellular viability after olaparib treatment. Strikingly, KAT5 or HUWE1 knockdown in HeLa cells depleted of BRCA2 using siRNA led to PARPi resistance similar to that seen in HR-proficient controls (Fig. 3a). This robust rescue of PARPi sensitivity was also observed in our HeLa BRCA2KO cells upon depletion of KAT5 (Supplementary Fig. 6a, b) or HUWE1 (Supplementary Fig. 6c, d) using either of two siRNA targeting sequences. We next asked whether these phenotypes were cell line-specific by extending our studies into additional cell lines. While KAT5 or HUWE1 depletion alone did not affect the sensitivity of wildtype controls to PARPi,

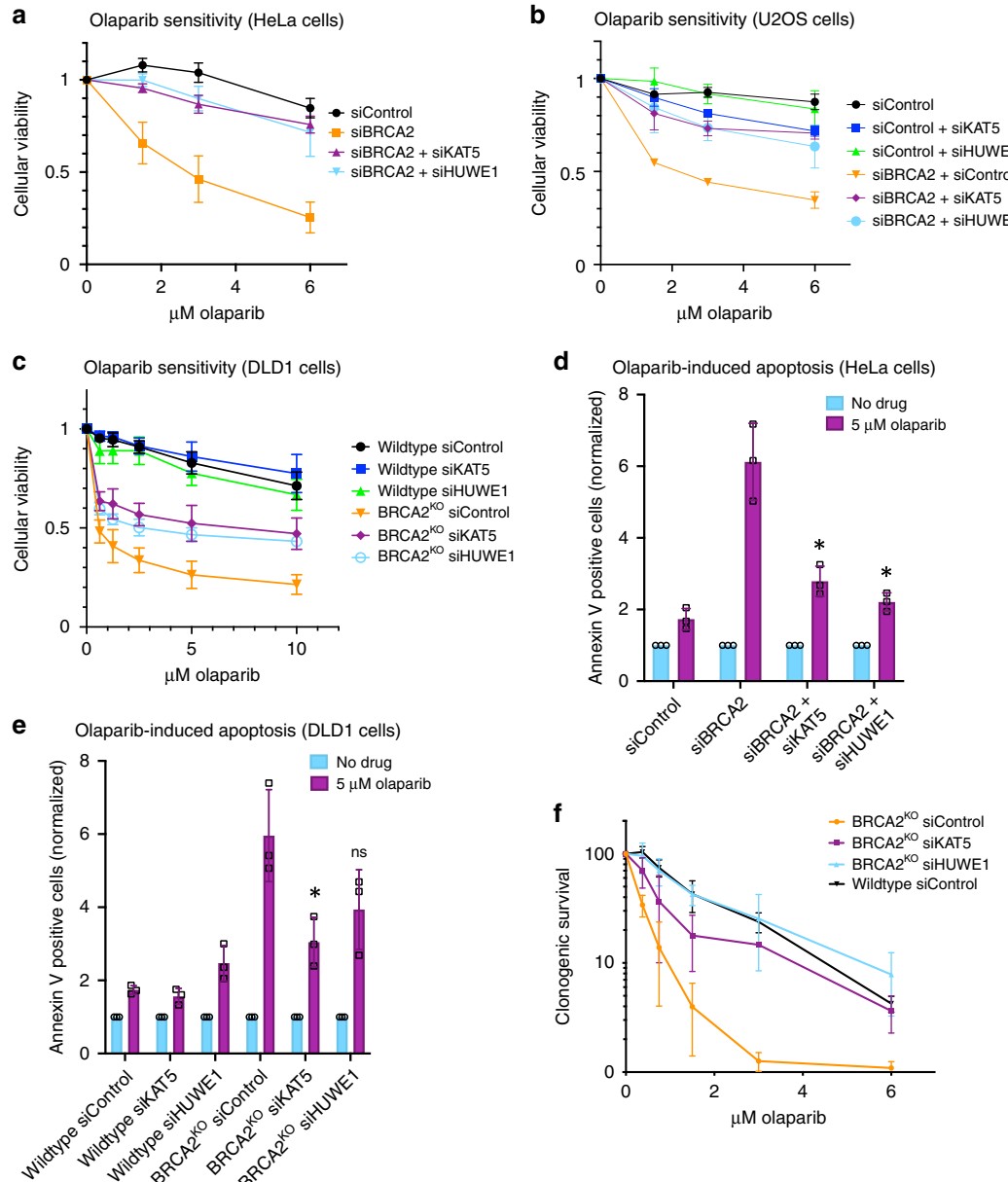

**Fig. 3 KAT5 or HUWE1 knockdown in BRCA2-depleted cells results in PARPi resistance. a–c** Knockdown of KAT5 or HUWE1 rescues the olaparib sensitivity of BRCA2-knockdown HeLa (**a**), BRCA2-knockdown U2OS (**b**), and BRCA2-knockout DLD1 (**c**) cells in cellular viability assays. **d**, **e** Olaparib-induced apoptosis is rescued by KAT5 or HUWE1 depletion in BRCA2-knockdown HeLa (**d**) and BRCA2-knockout DLD1 (**e**) cells. Asterisks indicate statistical significance (t-test, two-tailed, unequal variance), compared to siBRCA2 (**d**) or BRCA2$^{KO}$ siControl (**e**) samples. **f** KAT5 or HUWE1 depletion rescues sensitivity of BRCA2-knockout HeLa cells to PARPi in a clonogenic survival assay. All graphs reflect the average of three experiments performed after 72 h of treatment, with standard deviations shown as error bars. Source data are provided as a Source Data file.

co-depletion of either protein in BRCA2-depleted U2OS (Fig. 3b) or BRCA2-knockout DLD1 (Fig. 3c) cells significantly reduced PARPi sensitivity.

We next investigated the effect of KAT5 or HUWE1 depletion on olaparib-induced apoptosis using Annexin V flow cytometry. Consistent with the results of the cellular viability assays, depletion of KAT5 or HUWE1 abrogated the increase in Annexin V–positive cells caused by PARPi treatment in BRCA2-knockdown HeLa cells as well as BRCA2-knockout DLD1 cells (Fig. 3d, e). We observed similar phenotypes using a second siRNA oligonucleotide for each hit in HeLa BRCA2$^{KO}$ cells (Supplementary Fig. 6e, f).

In light of this increased cellular survival after olaparib treatment in BRCA2-deficient cells depleted of KAT5 or HUWE1, we wondered if these surviving cells would demonstrate improved viability in the long-term. To test this, we performed clonogenic survival assays in HeLa BRCA2$^{KO}$ cells. Cells were pre-treated with siRNAs targeting KAT5, HUWE1, or a control siRNA, treated with olaparib for three days, and then allowed to form colonies for two weeks in drug-free media. We found that while BRCA2$^{KO}$ cells showed reduced colony-forming ability after olaparib treatment, KAT5- and HUWE1-depleted BRCA2$^{KO}$ cells were able to form colonies after olaparib treatment in a manner similar to wildtype controls (Fig. 3f). Taken together, these findings confirm the results of the CRISPR knockout screen and demonstrate that KAT5 or HUWE1 depletion leads to PARPi resistance in BRCA2-deficient cells.

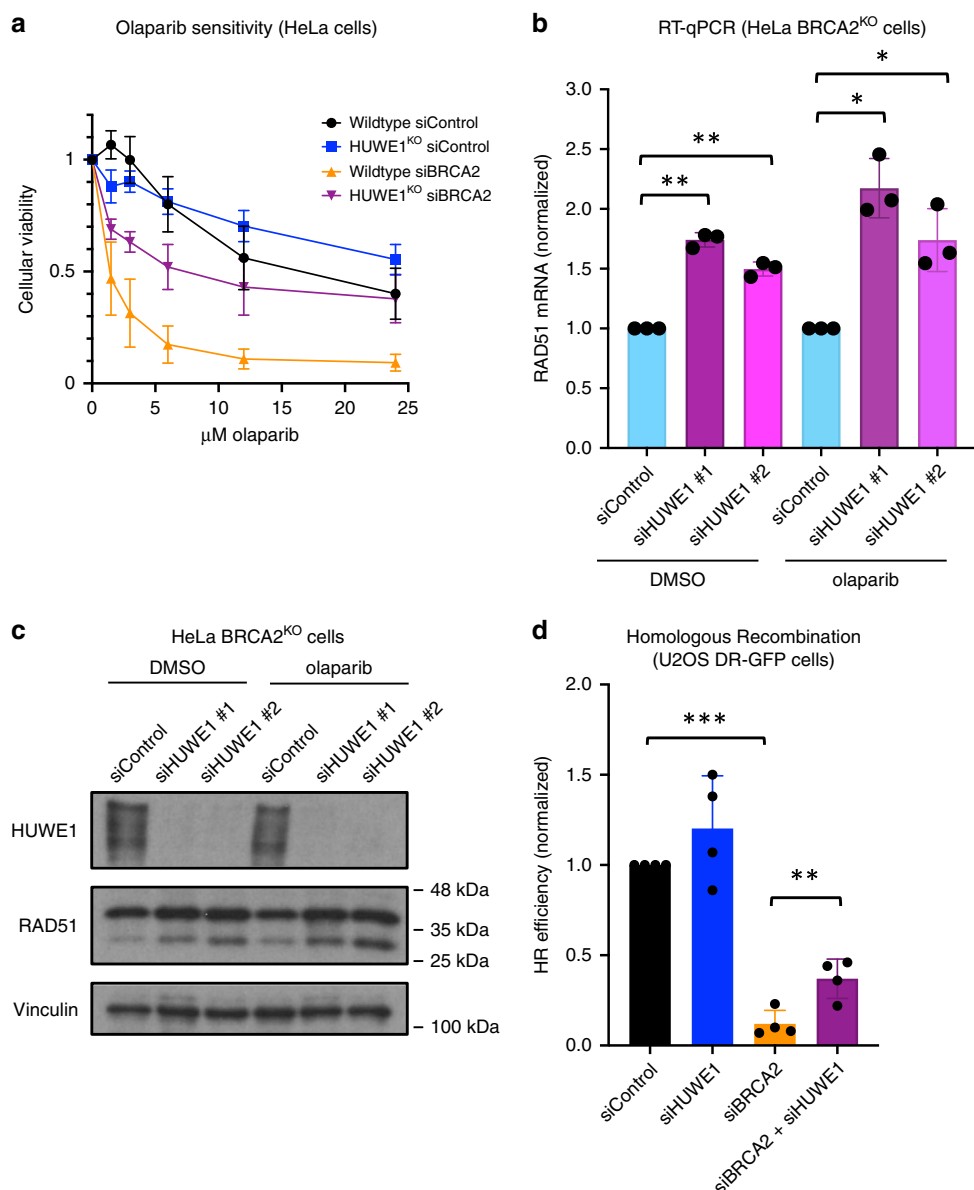

**Fig. 4 Loss of HUWE1 results in increased RAD51 levels and partial restoration of homologous recombination in BRCA2-deficient cells. a** Knockout of HUWE1 rescues the olaparib sensitivity of BRCA2-depleted HeLa cells in cellular viability assays. The graph presents the average of three experiments performed after 72 h of treatment, with standard deviations shown as error bars. **b** Quantitative PCR assay showing that HUWE1 depletion increases RAD51 mRNA levels. Cells were treated with DMSO or 10 µM olaparib for 24 h, as indicated. Data shown are the average of three experiments, normalized to the siControl condition, with standard deviations shown as error bars. Statistical significance is indicated by asterisks (*t*-test, two-tailed, unequal variance). **c** Western blot of whole-cell extracts showing that HUWE1 knockdown increases RAD51 protein levels. Cells were treated with DMSO or 10 µM olaparib for 24 h, as indicated. **d** HUWE1 depletion partially rescues the homologous recombination defect caused by BRCA2 knockdown as shown using a DR-GFP assay. The averages of four experiments are shown, with standard deviations as error bars. Statistical significance is indicated by asterisks (*t*-test, two-tailed, unequal variance). Source data are provided as a Source Data file.

**Loss of HUWE1 increases RAD51 levels and improves HR.** To validate the impact of HUWE1 loss on olaparib sensitivity of BRCA2-deficient cells using a complementary system, we employed HUWE1-knockout (HUWE1[KO]) HeLa cells[23]. Similar to our studies with HUWE1 depletion by siRNA, HUWE1 knockout increased PARPi resistance of BRCA2-knockdown cells, despite not affecting olaparib sensitivity in BRCA2-proficient cells (Fig. 4a). Moreover, BRCA2-knockdown HUWE1[KO] cells demonstrated significantly reduced olaparib-induced apoptosis as compared to BRCA2-knockdown wildtype cells (Supplementary Fig. 7a). These results further validate our

findings that HUWE1 depletion leads to PARPi resistance in BRCA2-deficient cells.

We next sought to explore this effect from a mechanistic standpoint. Interestingly, unbiased RNA-sequencing analyses of the transcriptional profile of HUWE1[KO] HeLa cells showed a ~50% increase in the expression of RAD51 mRNA (Supplementary Fig. 7b). In order to understand if this is a direct result of loss of HUWE1, we knocked-down HUWE1 using multiple siRNA oligonucleotides and measured RAD51 levels. RT-qPCR analyses confirmed that HUWE1 loss results in an increase in RAD51 mRNA (Fig. 4b). Importantly, this was associated with a

corresponding increase in RAD51 protein levels (Fig. 4c). We previously showed that increasing RAD51 expression by 50% enhances PARPi resistance in BRCA2-deficient cells, by partially rescuing the defect in HR[14]. To explore this possibility, we employed the DR-GFP HR assay[24]. Knockdown of HUWE1 alone did not significantly impact HR, while BRCA2 knockdown, as expected, significantly suppressed it (Fig. 4d). Importantly, HUWE1 co-depletion resulted in a significant increase in HR efficiency of BRCA2-deficient cells. Overall, these findings suggest that loss of HUWE1 may potentially promote olaparib resistance in BRCA2-deficient cells by partially restoring HR via an increase in RAD51 levels.

**PARPi resistance via KAT5 depletion requires 53BP1 and REV7**. After observing that KAT5 depletion robustly rescued olaparib sensitivity, we investigated if this also applies to other PARP inhibitors. To this end, we depleted KAT5 from BRCA2-knockout cells and treated them with veliparib or niraparib. In both cases, we observed that loss of KAT5 restores resistance to these PARP inhibitors (Supplementary Fig. 8a, b), indicating that this response is not specific to olaparib but instead applies to other PARP inhibitors as well.

Next, we sought to investigate the mechanisms through which this may occur. To this end, we tested the effect of KAT5 depletion on several previously proposed mechanisms of PARPi-induced cytotoxicity. Trapping of PARP1 on the chromatin has been suggested as an underlying cause of the toxicity of PARPi[17]. Given that KAT5 has been previously connected to the flux of PARP1 on chromatin[21], we tested if KAT5 depletion may abrogate this PARP-trapping effect of the inhibitor. However, we observed no difference in trapped PARP1 after KAT5 depletion using a chromatin fractionation assay (Fig. 5a, Supplementary Fig. 9a). Furthermore, an aberrant increase in replication fork speed was recently proposed as a mechanism of action of PARPi[25]. While we did observe an increase in replication fork speed upon PARPi treatment, this phenotype was not affected by KAT5 depletion (Supplementary Fig. 9b). In contrast to HUWE1, KAT5 depletion did not improve the HR efficiency of BRCA2-depleted cells (Supplementary Fig. 9c). Interestingly, however, despite no change in HR repair, we observed a reduction in olaparib-induced double-strand breaks in HeLa BRCA2[KO] cells after KAT5 depletion using a neutral comet assay (Fig. 5b), suggesting that repair of DSBs is improved in these cells.

Previous studies have indicated that KAT5 activity antagonizes the recruitment of 53BP1 to damaged DNA[26–28]. 53BP1 is a key mediator of double-strand break repair pathway choice, which suppresses DNA end resection at DSBs and promotes non-homologous end joining (NHEJ)[29,30]. We reasoned that increased 53BP1 binding due to KAT5 depletion may be involved in the improved cellular viability and reduced double-strand breaks observed after olaparib treatment in BRCA2-deficient cells. To test if 53BP1 is required for PARPi resistance in BRCA2-deficient cells depleted of KAT5, we performed additional cellular viability assays. KAT5 was depleted alone or in combination with 53BP1 in HeLa BRCA2[KO] cells and sensitivity to PARPi was assessed. We found that while KAT5 depletion alone causes resistance, co-depletion of 53BP1 with KAT5 severely diminishes this rescue (Fig. 5c). Importantly, the magnitude of KAT5 knockdown was unaffected by 53BP1 co-depletion (Supplementary Fig. 10a). Recently, the Shieldin complex has been identified as a downstream effector of 53BP1 at the double-strand break[31,32]. Therefore, we next tested if REV7, a component of the Shieldin complex, was also required for the rescue of PARPi sensitivity caused by KAT5 loss. Indeed, co-depletion of REV7 abolished the PARPi resistance produced by the depletion of KAT5 (Fig. 5d,

Supplementary Fig. 10b). Taken together, these results suggest that the 53BP1/REV7 pathway is required for PARPi resistance caused by KAT5 depletion in BRCA2-deficient cells.

**KAT5 regulates 53BP1 binding and end resection at DSBs**. We next sought to directly assess the impact of KAT5 depletion on the binding of 53BP1 near the double-strand break site in BRCA2-deficient cells. We used a chromatin immunoprecipitation (ChIP) assay with a 53BP1 antibody to evaluate binding near the site of an inducible double-strand break within an integrated reporter in the previously described U2OS-DSB reporter cell line[27]. qPCR analysis of the immunoprecipitated DNA revealed increased 53BP1 binding near the double-strand break site after KAT5 knockdown in BRCA2-depleted cells, as measured 5 h after DSB induction (Fig. 6a). Moreover, immunofluorescence analyses indicated that while there was no difference in 53BP1 intensity at DSB sites between genetic conditions at the 1-h post-induction timepoint, at 2.5 h after treatment there was a trend towards greater 53BP1 accumulation in KAT5-depleted cells (Supplementary Fig. 11a). At both timepoints, cells depleted of KAT5-showed an increase in γH2AX intensity (Supplementary Fig. 11b), consistent with previous reports that KAT5 depletion leads to increased retention of γH2AX at sites of damage[33–35]. However, both control and KAT5-depleted cells showed similar 1.25-fold increases in γH2AX from the 1-h to the 2.5-h timepoint, indicating similar kinetics of DNA repair within the timeframe in which 53BP1 is beginning to accumulate. These findings suggest that the increase in 53BP1 binding at DSBs upon KAT5 depletion occurs gradually over time, and is not caused by a delay in DNA repair.

As the 53BP1/Shieldin pathway functions to prevent excessive-end resection at DNA ends, we sought to investigate the effect of KAT5 depletion on end resection[31]. To test this, we quantified the percentage of cells that were positive for RPA, a protein which binds to single-stranded DNA produced by end resection, using flow cytometry[36,37]. We observed an accumulation of S-phase cells in control cells treated with olaparib, which was not observed in siKAT5-treated cells (Supplementary Fig. 12). To prevent these differences from confounding our analysis, we only assessed RPA-positivity in G2 and M-phase cells. In this population of cells, olaparib treatment caused an increase in the percentage of RPA-positive cells, which was reduced upon KAT5 depletion (Fig. 6b). Overall, these findings suggest that KAT5 depletion increases 53BP1 binding to olaparib-induced DSBs in BRCA2-deficient cells, suppressing end resection.

Altogether, we observed that KAT5 depletion causes PARPi resistance in a manner potentially dependent on the 53BP1/REV7 pathway, and subsequently causes a reduction in olaparib-induced end resection. Notably, inhibiting end resection directly via depletion of CTIP was not sufficient to cause PARPi resistance in BRCA2-deficient cells (Supplementary Fig. 13a, b). Previously, exhaustion of 53BP1 by an abundance of DSBs (as one might expect in BRCA2-deficient cells treated with PARPi) was shown to lead to an increase in single strand annealing (SSA), a repair pathway downstream of end resection that results in large deletions and genomic instability[38]. Therefore, we reasoned that the observed increase in 53BP1 binding may rescue PARPi sensitivity by preventing SSA. However, inhibiting SSA via depletion of LIG1 also failed to cause PARPi resistance in this context (Supplementary Fig. 13a, b). One potential explanation for these findings is that increased 53BP1 recruitment not only reduces end resection, but also promotes DSB repair through NHEJ[29]. To address this, we investigated NHEJ DNA repair signature at somatic structural variation breakpoints in publicly available genomic datasets. Bioinformatic analysis of the

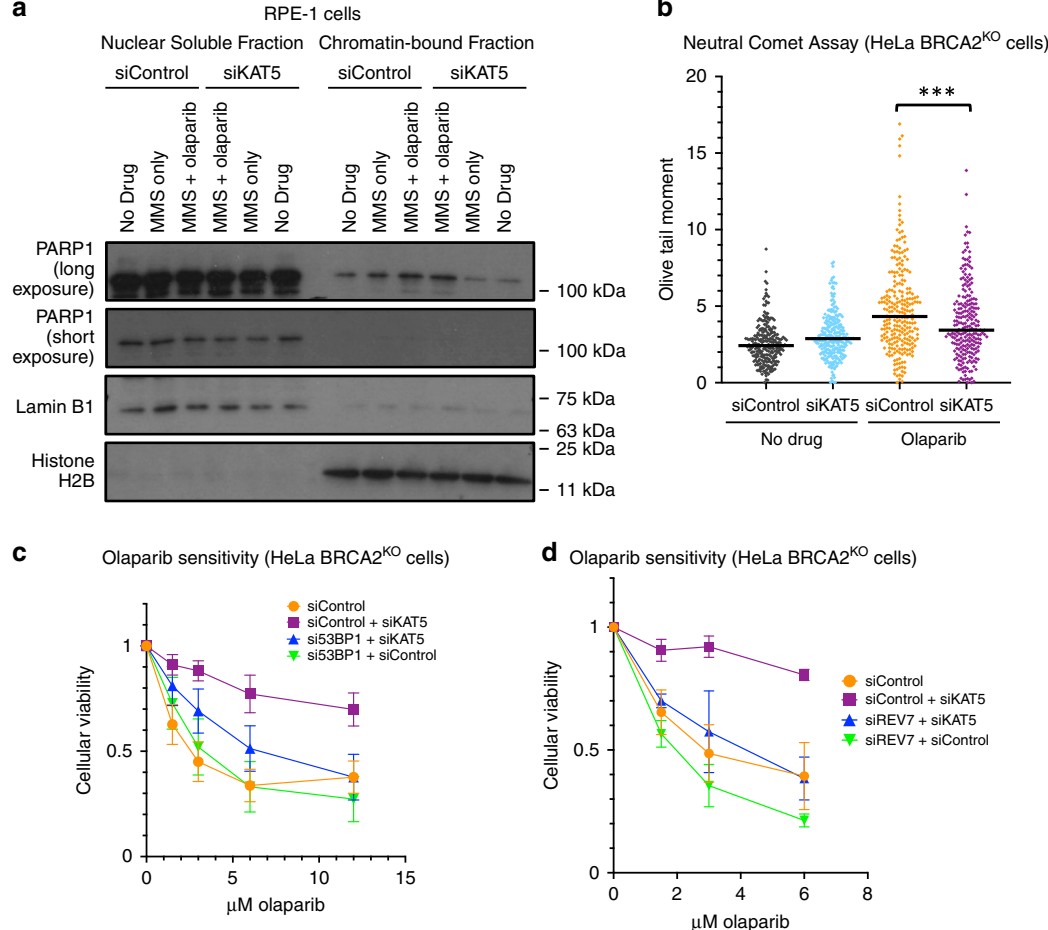

**Fig. 5 KAT5 depletion reduces olaparib-induced double-strand breaks and relies on the 53BP1/REV7 pathway to rescue olaparib-induced cytotoxicity.**
**a** Chromatin fractionation experiment shows that while MMS (0.01%) and olaparib (1 μM) co-treatment for 3 h induces trapping of PARP1 on the chromatin, this effect is not rescued by KAT5 depletion. An independent replicate experiment is shown in Supplementary Fig. 9a. **b** Neutral comet assay showing that olaparib treatment (10 μM for 24 h) induces double-strand breaks in HeLa BRCA2-knockout cells, and that this effect is abrogated by KAT5 depletion. At least 120 comets from each of two experiments were pooled for each sample. Medians are shown as horizontal bars. Asterisks indicate statistical significance (Mann–Whitney test, two-sided). **c** PARPi resistance caused by KAT5 depletion is dependent on 53BP1. The averages of nine experiments are shown, with standard deviations as error bars. **d** REV7 is required for the PARPi resistance after KAT5 knockdown. The averages of three experiments are shown, with standard deviations as error bars. Source data are provided as a Source Data file.

Australian Ovarian Cancer Study cohort (OV-AU) dataset suggests a higher incidence of NHEJ in ovarian tumors with low KAT5 expression, in both BRCA2-high and BRCA2-low backgrounds (Fig. 6c). Overall, these findings suggest a model in which KAT5 depletion increases 53BP1 binding near the double-strand break, leading to a reduction in end resection and subsequent PARPi resistance, potentially through an increase in NHEJ (Fig. 6d).

**KAT5 levels impact cisplatin resistance.** Several factors which confer resistance to PARPi have also been shown to rescue sensitivity to the DNA damaging agent and widely used chemotherapeutic, cisplatin[39]. Thus, we reasoned that KAT5 depletion may also confer resistance to cisplatin in BRCA2-deficient cells. Using a clonogenic survival assay, we found that depletion of KAT5 in BRCA2-knockout HeLa cells rescues cisplatin sensitivity back to that of wildtype cells (Fig. 7a). Moreover, nucleolytic degradation of stalled replication forks by the nuclease MRE11 is a hallmark of BRCA-deficient cells[40], and cisplatin resistance was previously correlated with increased protection

against MRE11 activity[8,9]. In line with this, we observed that KAT5 depletion also rescued the fork degradation defect of BRCA2-deficient cells (Fig. 7b).

Unlike PARPi, which are relatively new agents in the clinic, cisplatin has remained a mainstay of ovarian cancer treatment for decades[41]. Thus, we next sought to investigate if KAT5 expression impacts survival of BRCA2-mutant ovarian cancer patients by mining survival data and matched genotype and expression data from publicly available datasets. We queried the OV-TCGA database and asked if ovarian cancer patients with low or high KAT5 expression showed any differences in survival, in BRCA2-mutant or BRCA2-wildtype backgrounds. We found that low KAT5 expression in tumors trended towards poorer survival in patients harboring BRCA2 mutations, but not in the BRCA2-wildtype cohort (Fig. 7c). As it is highly likely that most patients in this dataset have been treated with cisplatin, these findings are potentially consistent with increased therapy resistance in the KAT5-low BRCA2 mutant group. Taken together, these data indicate that KAT5 depletion confers resistance to cisplatin in BRCA2-deficient cells and potentially also in tumors.

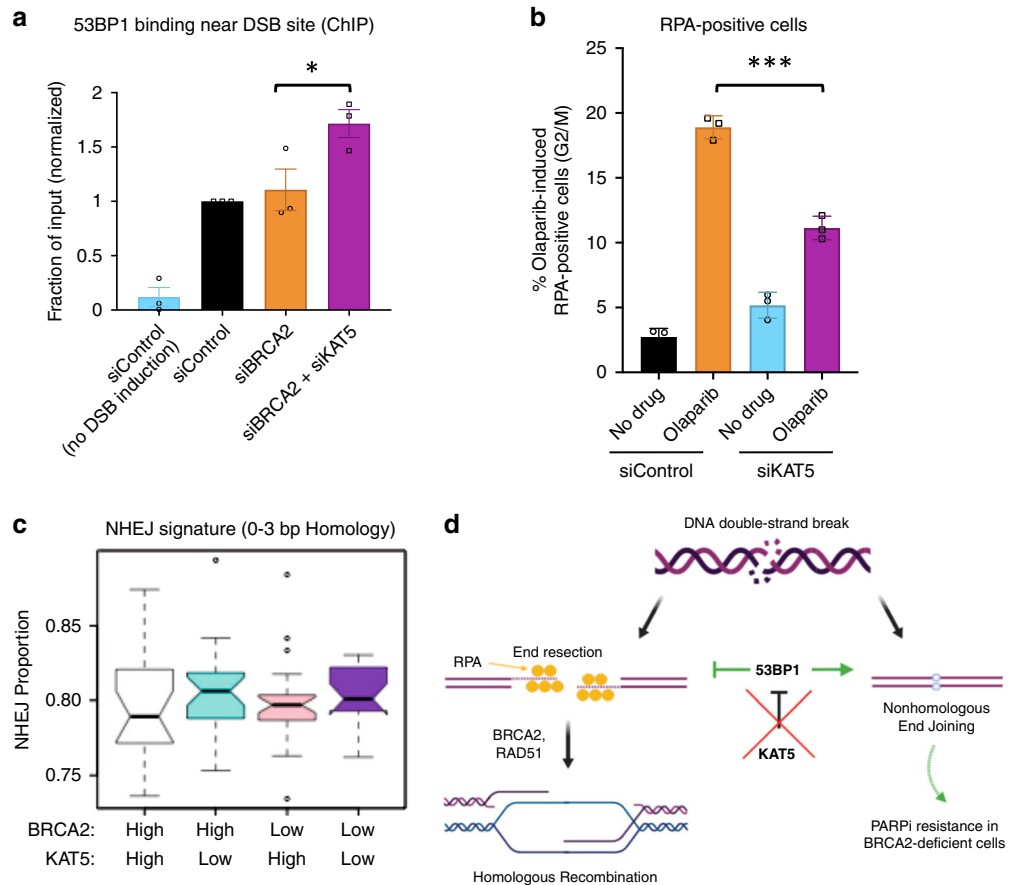

**Fig. 6 Functional consequences of KAT5 depletion. a** Depletion of KAT5 leads to increased recruitment of 53BP1 near the site of the double-strand break in BRCA2-depleted U2OS-DSB reporter cells. DSBs were generated at a specific genomic locus through induction of expression of LacI-FokI nuclease. Then, chromatin immunoprecipitation was performed with an antibody against 53BP1 and qPCR was performed using primers for the DSB site. The averages of three experiments are shown, with standard error of the mean as error bars. Asterisks indicate statistical significance (*t*-test, two-tailed, paired). **b** KAT5 loss leads to a reduction in olaparib-induced end resection as measured by quantification of RPA-positive cells. Cells were treated with 10 μM olaparib for 12 h before being harvested for flow cytometry experiments using an RPA antibody. Representative gating of these experiments is shown in Supplementary Fig. 12. The averages of three experiments are shown, with standard deviations as error bars. Asterisks indicate statistical significance (*t*-test, two-tailed, unequal variance). **c** Bioinformatic analysis of the OV-AU dataset reveals that KAT5 low tumors show increased NHEJ signature at structural variations breakpoints, as compared to KAT5 high tumors, in both BRCA2 high and BRCA2 low backgrounds. The 93 samples in the cohort were distributed into four categories based on combinatorial high (above median) and low (below median) expression of BRCA2 and KAT5. In the box plots, lower and upper hinges correspond to the first and third quartiles; upper and lower whiskers extend to 1.5 × interquartile range; the line in the box indicates the median; outlier data are shown as points. **d** Proposed model for resistance caused by KAT5 depletion. In BRCA2-proficient cells, homologous recombination is the most accurate mechanism of double-strand break repair. However, this pathway is not functional in BRCA2-deficient cells. KAT5 depletion increases 53BP1 binding at the double-strand break in BRCA2-deficient cells, preventing the break from proceeding down the dead-end process of HR and instead directs it towards repair by NHEJ. Source data are provided as a Source Data file.

## Discussion

Identifying potential predictors of response to PARPi serves to both identify additional patients who may benefit from this therapy, and to avoid ineffective treatment for those who will not. In addition, beyond simply predicting treatment response, achieving a better understanding of mechanisms of sensitivity and resistance to PARPi brings us closer to potential therapeutic interventions that may be able to sensitize or re-sensitize patients to treatment. Therefore, there has been great interest in investigating PARPi sensitivity and resistance through genome-wide CRISPR knockout screens in cells with various genetic backgrounds[13,39,42].

Here, we have completed a series of genome-wide screens based on CRISPR technology as an unbiased approach towards better understanding determinants of PARPi response. First, we present data from a CRISPR-knockout screen in wildtype HeLa cells, which identified genes which, when depleted, lead to PARPi sensitivity. In addition, we investigated PARPi resistance in BRCA2-knockout HeLa cells by identifying genes which cause resistance when depleted (CRISPR knockout library) or over-expressed (CRISPR activation library). By utilizing two different types of libraries and otherwise isogenic cell lines differing only in BRCA2 levels, we are able to not only examine the results of each screen individually, but also to consider the results in relationship to one another. For example, loss of POLE3 or POLE4, subunits of DNA polymerase Epsilon, sensitized parental HeLa cells to PARPi treatment, while transcriptional activation of polymerase Epsilon subunit POLE2 was associated with PARPi resistance in BRCA2-knockout HeLa cells. This is especially interesting given that depletion of polymerase Epsilon subunits was recently shown to lead to sensitivity to ATR inhibitors[43].

A potential limitation of our study is the use of HeLa cells for the genome-wide screens, since hits specific to breast, ovarian, or pancreatic cells—cancers for which PARPi are currently clinically

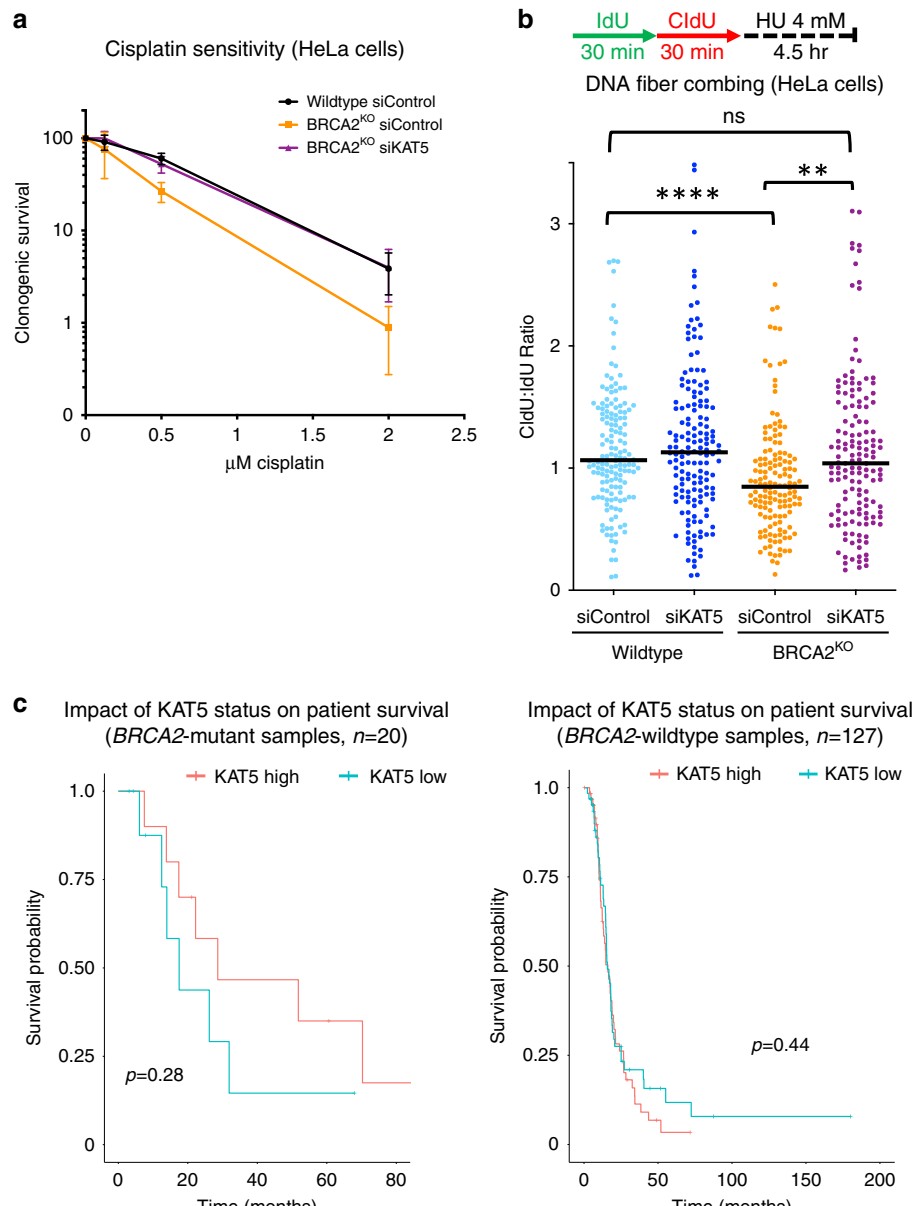

**Fig. 7 Loss of KAT5 expression causes resistance to cisplatin in BRCA2-deficient cells. a** In clonogenic survival assays, depletion of KAT5 in BRCA2-knockout HeLa cells led to increased colony formation after cisplatin treatment (24 h). The averages of three experiments are shown, with standard deviations as error bars. **b** DNA fiber combing assay showing that KAT5 depletion partially suppresses the HU-induced nascent DNA degradation in BRCA2-deficient cells. The ratio of CldU to IdU tract lengths is presented, with the median values marked on the graph. At least 150 replication tracts were quantified for each sample. Asterisks indicate statistical significance (Mann–Whitney test, two-sided). **c** Bioinformatic analysis of the OV-TCGA dataset shows that patients with tumors that have low KAT5 expression trend towards poorer survival in BRCA2 mutant (left) but not BRCA2 wildtype (right) tumors. The difference in BRCA2 mutant samples is not significant ($p = 0.28$) likely because of the small number ($n = 20$) of BRCA2-mutant samples in this dataset. Source data are provided as a Source Data file.

indicated—may have been missed. In addition, the experimental setup (4-day drug treatment as opposed to a longer treatment time with a lower dose) may have limited the scope of the relevant hits identified by our studies.

Our results identified several hits which were consistent with previous findings from other groups, as well as novel hits which have not previously been connected to PARPi response. One such novel hit is HUWE1, which we validated using several cellular survival and apoptosis assays, in multiple cell lines. Mechanistically, we show that loss of HUWE1 results in increased RAD51 levels and partial restoration of HR, in line with previous reports that increasing RAD51 expression can promote olaparib

resistance in BRCA2-deficient cells[14]. HUWE1 is a ubiquitin ligase with many known targets[44]. At this time, it is unknown which HUWE1 targets, if any, may be implicated in the phenotypes described here. Interestingly, computational analyses of our RNA-sequencing studies showed that gene networks associated with a number of RAD51 transcriptional regulators—including MYC, YY1, E2F1, E2F4, and E2F6—are differentially expressed in HUWE1-knockout cells compared to wildtype cells. This raises the possibility that HUWE1 may directly ubiquitinate some of these factors, thus providing a mechanistic link to the observed phenotypes. Indeed, HUWE1 was previously shown to ubiquitinate MYC[45].

Another novel hit which we validated using multiple assays in multiple cell lines is KAT5. Our results argue that the PARPi resistance caused by KAT5 depletion is dependent on both 53BP1 and REV7, which have previously been shown to dissociate from DNA upon KAT5-mediated acetylation of histones H4 and H2A[26–28,31,32]. Functionally, we show that KAT5 depletion increases the binding of 53BP1 near the double-strand break, reduces olaparib-induced end resection, and ameliorates the olaparib-induced increase in double-strand breaks. This potentially represents a novel mechanism of PARPi resistance in BRCA2-deficient cells, highlighting critical differences between BRCA2- and BRCA1-deficiency. In fact, whereas we show that in BRCA2-deficient cells an increase in 53BP1 binding to damaged DNA plays a role in PARPi resistance, others have shown 53BP1 loss to be a mechanism of PARPi resistance in BRCA1-deficent cells[30,46]. Although we have not directly tested the effect of KAT5 depletion on PARPi sensitivity of BRCA1-deficient cells, the apparently divergent impact of 53BP1 on PARPi resistance in BRCA2- and BRCA1-deficient cells possibly reflects the different roles of BRCA1/2 in the double-strand break repair process of HR, and is consistent with previous reports that 53BP1 depletion rescued defects in proliferation and checkpoint responses in BRCA1-deficient, but not BRCA2-deficient cells[47]. BRCA1 functions early in HR, promoting end resection at the double-strand break; this is in direct opposition to the anti-resection role of 53BP1[29]. Therefore, 53BP1 depletion would be expected to alleviate the defects caused by BRCA1 deficiency and improve the ability of the cell to manage olaparib-induced damage, resulting in resistance. On the other hand, BRCA2 functions much later in the process of HR. Hence, 53BP1 depletion would not correct this defect; instead, increased 53BP1 binding would reduce end resection and prevent the break from beginning down the HR pathway, which is a dead-end in BRCA2-deficient cells. Altogether, our work uncovers an unexpected mechanism of PARPi resistance in BRCA2-deficient cells and argues that despite both causing defects in HR, BRCA1- and BRCA2- deficiency represent distinct defects that should not always be grouped together in the laboratory or the clinic.

Our work suggests that 53BP1 may be required for the rescue of PARPi sensitivity caused by KAT5 depletion, and also that KAT5-deficient patient tumors show an increase in the non-homologous end-joining (NHEJ) pathway of double-strand break repair, consistent with the expected effects of an increase in 53BP1 binding at the double-strand break[29,30]. However, 53BP1 has been implicated in additional PARPi-relevant contexts, such as at the replication fork[48–50], and here, we also show that KAT5 depletion partially rescues the fork protection defect of BRCA2-deficient cells. While there is a clear correlation between fork protection and chemoresistance, the exact contribution of fork protection to olaparib resistance is unclear[6,9,51]. In addition, the finding that REV7 is also required for the resistance caused by KAT5 depletion supports a mechanism involving activity at the double-strand break[52]; however, other potential roles for REV7 at the replication fork have not yet been thoroughly characterized[53]. Therefore, other function(s) of 53BP1/REV7 may potentially be involved in the rescue of PARPi-induced cytotoxicity downstream of KAT5 depletion in BRCA2-deficient cells.

## Methods

**Cell culture**. Human HeLa, U2OS, and RPE-1 cells were grown in Dulbecco's modified Eagle's medium (DMEM). DLD1 cells were grown in Roswell Park Memorial Institute (RPMI) 1640 medium. DMEM and RPMI were supplemented with 10% fetal bovine serum. HeLa BRCA2[KO] and HeLa HUWE1[KO] cells were generated in our laboratory[14,23]. DLD1 BRCA2[KO] cells (Horizon HD105-007) were obtained from Dr. Robert Brosh (National Institute on Aging, NIH). U2OS-DSB reporter cells were obtained from Dr. Roger Greenberg (University of Pennsylvania)[27]. U2OS DR-GFP cells were obtained from Dr. Jeremy Stark (City of Hope National Medical Center, Duarte, CA)[24].

Gene knockdowns were performed using Lipofectamine RNAiMAX reagent for transfection of Stealth siRNA (Life Tech, unless otherwise noted). AllStars Negative Control siRNA (Qiagen 1027281) was used as control. Oligonucleotide sequences used were: BRCA2: GAGAGGGCCTGTAAAGACCTTGAATT; KAT5 #1: GATGG ACGTAAGAACAAGAGTTATT; KAT5 #2: CACCCATTCATCCAGACGTTTGT TG; HUWE1 #1: TTTAAGGGTGGGCTGATGTCTCATG; HUWE1 #2: CACACC AGCAATGGCTGCCAGAATT; 53BP1: TCCCAGAGTTGATGTTTCTTGTGAA; REV7: GTGGAAGAGCGCGCTCATAAA (Qiagen); CTIP: GGGTCTGAAGTG AACAAGATCATTA; LIG1: CCAAGAACAATCATCCCGTGGA. For all genes, siRNA #1 was used for knockdown unless otherwise indicated.

Knockdown was confirmed by western blot for all proteins targeted, except REV7, which was assessed using real-time quantitative PCR (RT-qPCR)[14]. Total mRNA was purified using TRIzol (Life Tech). To generate cDNA, 1 µg RNA was subjected to reverse transcription using the RevertAid Reverse Transcriptase Kit (Thermo Fisher Scientific) with oligo-dT primers. Real-time qPCR was performed with PerfeCTa SYBR Green SuperMix (Quanta), using a CFX Connect Real-Time Cycler (BioRad). The cDNA of GAPDH gene was used for normalization. Primers used were: REV7 for: TGCTGTCCATCAGCTCAGAC; REV7 rev: TCTTCTCCA TGTTGCGAGTG; GAPDH for: AAATCAAGTGGGGCGATGCTG; GAPDH rev: GCAGAGATGATGACCCTTTTG. RAD51 expression was also assessed by RT-qPCR; primers used were: RAD51 for: TGCTTATTGTAGACAGTGCCACC; RAD51 rev: CACCAAACTCATCAGCGAGTC.

HeLa BRCA2-knockout cells overexpressing ABCB1 via transcriptional activation were created by consecutive rounds of transduction and selection. Cells were first infected with dCas9 (Addgene, 61425-LV) and colonies were selected with blasticidin, 3 µg/ml. dCas9-expressing cells were then infected with the MS2-P65-HSF1 activator helper complex (Addgene, 61426-LVC) and treated with 0.5 mg/ml hygromycin. Finally, hygromycin-resistant cells were infected with lentivirus containing sgRNA targeting ABCB1 (Sigma Custom CRISPR in lentiviral backbone LV06), guide sequence (5'-3'): GGGAGCAGTCATCTGTGGTG. Infected cells were selected using puromycin (0.6 µg/ml).

**Genome-wide CRISPR screens**. For the CRISPR knockout screens, wildtype or BRCA2-knockout HeLa cells were transduced with the Brunello Human CRISPR knockout pooled library (Addgene, 73179)[10]. To achieve a representation of 250 cells per sgRNA, 50 million cells were transduced at a low multiplicity of infection (MOI) (0.4). Selection with puromycin (0.6 µg/ml) was initiated at 48 h post-transduction and maintained for 4–6 days. For the resistance screen in BRCA2-deficient cells, puromycin-resistant cells were divided into olaparib- or vehicle-(DMSO) treated arms at 250-fold library coverage per arm. Olaparib was used at 4 µM for 4 days, and yielded 7 and 10% survival relative to the DMSO condition in the two screen replicates. After treatment, cells were pelleted and flash-frozen for DNA extraction. For the sensitivity screen in wildtype cells, puromycin-resistant cells were seeded at a representation of 150 cells per sgRNA and treated with DMSO or olaparib (5 µM) for 4 days. After the 4 days of treatment, the survival of the olaparib-treated population was 43 and 46% of the vehicle-treated cells in the two screen replicates; cells were pelleted and flash-frozen for DNA extraction.

For the CRISPR activation screens, HeLa BRCA2-knockout cells were infected with dCas9 (Addgene, 61425-LV) and selected with blasticidin (3 µg/ml). dCas9-expressing cells were then transduced with the Calabrese Human CRISPR Activation Pooled Library (Set A, Addgene, 92379-LV) using enough cells to obtain a library coverage of 500 cells per sgRNA at an MOI of 0.4[18]. Selection with puromycin (0.6 µg/ml) was initiated at 48 h post-transduction and maintained for 5 days. After puromycin selection, cells were divided into olaparib-(4 µM) or vehicle-(DMSO) treated arms at 500-fold library coverage per arm. After 4 days of treatment, the survival of the olaparib-treated cells in the two screen replicates was 12 and 16% relative to the DMSO-treated condition; cells were flash-frozen as pellets for DNA extraction.

**Sequencing and analysis of CRISPR screens**. Genomic DNA (gDNA) was extracted per manufacturer's instructions using the DNeasy Blood & Tissue Kit (Qiagen, 69504). A maximum of 5 million cells were used per column. Isolated gDNA was quantified using Nanodrop. The gDNA equivalent of 125-fold library coverage (knockout screens) or 250-fold library coverage (activation screen) was used as template for PCR amplification of the sgRNA sequences. Each PCR reaction included no more than 10 µg of gDNA, in addition to the following components: 3 µl of Radiant HiFi Ultra Polymerase (Stellar Scientific, RAD-HF1100), 20 µl of 5X HiFi Ultra Reaction Buffer, 4 µl of 10 µM P5 primer, 4 µl of 10 µM uniquely barcoded P7 primer, and water to bring the total reaction volume to 100 µl. Primers were synthesized by Eurofins Genomics using the sequences listed in the user guide provided for the CRISPR libraries. PCR cycling conditions were as follows: an initial 2 min at 98 °C; followed by 10 s at 98 °C, 15 s at 60 °C, 45 s at 72 °C, for 30 cycles; and lastly 5 min at 72 °C[54]. The E.Z.N.A. Cycle Pure Kit (Omega, D6493-02) was used per manufacturer's instructions to purify PCR products. PCR products were further purified with Agencourt AMPure XP SPRI beads according to manufacturer's instructions (Beckman Coulter, A63880). The final product was assessed for its size distribution and concentration using

BioAnalyzer High Sensitivity DNA Kit (Agilent Technologies) and qPCR (Kapa Biosystems). Pooled libraries were diluted to 2 nM in EB buffer (Qiagen) and then denatured using the Illumina protocol. The denatured libraries were diluted to 10 pM by pre-chilled hybridization buffer and loaded onto a TruSeq v2 Rapid flow cell on an Illumina HiSeq 2500 and run for 50 cycles using a single-read recipe according to the manufacturer's instructions (Illumina).

For bioinformatic analysis, de-multiplexed and adapter-trimmed sequencing reads were generated using Illumina bcl2fastq (released version 2.18.0.12) allowing no mismatches in the index read. Gene rankings for each screen were generated using MAGeCK (version 0.5.6)[11]. The MAGeCK algorithm allows the combined analysis of multiple replicates of each screen to generate a single ranking. First, the sequencing fastq files were converted to sgRNA read count tables by using the Count sub-command in MAGeCK. Since each screen was done in duplicate, the individual fastq files were inputted separated by a comma to denote replicates. Next, the MAGECK *test* sub-command was employed, which uses the robust ranking aggregation (RRA) algorithm to compare the sgRNA read counts between treatment and control conditions and generate sgRNA and gene scores. Genes were ranked by the negative selection score for the CRISPR knockout olaparib sensitivity screen in wildtype cells, and by the positive selection score for the CRISPR knockout and activation olaparib resistance screens in BRCA2$^{KO}$ cells.

**Protein techniques**. Total cellular protein extracts were prepared by boiling cells in 100 mM Tris, 4% SDS, 0.5 M β-mercaptoethanol[55]. For the PARP-trapping assay, cells were co-treated with MMS (0.01%) and olaparib (1 μM) for 3 h to induce trapping of PARP1[17]. Cellular fractionation was performed with the Subcellular Protein Fractionation Kit for Cultured Cells (Thermo Scientific, 78840) per manufacturer's instructions; protein was quantified using the Qubit Protein Assay Kit (Invitrogen). Antibodies used were: ABCB1 (Santa Cruz Biotechnology, sc-13131), GAPDH (Santa Cruz Biotechnology, sc-47724), KAT5 (Santa Cruz Biotechnology, sc-166323), HUWE1 (Bethyl, A300-486A), Vinculin (Santa Cruz Biotechnology, sc-25336), 53BP1 (Bethyl, A300-272A), CTIP (Santa Cruz Biotechnology, sc-271339), LIG1 (Bethyl, A301-136A), PARP1 (Cell Signaling, 9542), PCNA (Cell Signaling, 2586), Cas9 (BioLegend 844302), Histone H2B (Cell Signaling, 12364), Lamin B1 (Abcam, ab16048), RAD51 (Santa Cruz Biotechnology, sc-8349). All antibodies were used at a dilution of 1:500. Uncropped scans of all blots, including molecular weight markers, are provided as a Source Data file.

**Drug sensitivity assays**. Olaparib, veliparib, and niraparib were obtained from Selleck Chemicals and cisplatin was obtained from Biovision. To test cellular viability after olaparib treatment, cells were seeded in 96-well plates at a density of 2000 cells per well, treated with the indicated doses of olaparib for 3 days, and assessed using CellTiter-Glo reagent (Promega, G7572) per manufacturer's instructions. Clonogenic survival assays were performed by seeding cells in 6-well plates; after 72 h (olaparib) or 24 h (cisplatin) of treatment at the indicated concentrations, media was changed and colonies were allowed to form for 2 weeks. Cells were fixed with a solution of 10% methanol + 10% acetic acid and stained using crystal violet (2% solution, Aqua Solutions). For apoptosis assays, cells were treated with olaparib (5 μM) for 3 days, prepared for flow cytometry using the FITC Annexin V kit (Biolegend, 640906) and quantified using a BD FACSCanto 10 flow cytometer operated by BD FACSDiva 8.0.1 software.

**Functional assays**. The ChIP experiments to investigate 53BP1 binding near a double-strand break site were performed using the U2OS-DSB cell line with an integrated reporter transgene and inducible expression of mCherry-LacI-FokI nuclease[27]. Cells were pretreated with the indicated siRNA for 48 h, then treated with 4-hydroxytamoxifen (4-OHT, 1 μM) and shield-1 ligand (1 μM) to induce nuclease expression and subsequent double-strand break formation. Five hours after induction, cells were harvested and processed using the SimpleChIP Enzymatic Chromatin Immunoprecipitation Kit (Cell Signaling, 9003) according to the manufacturer's instructions. The antibody used for immunoprecipitation was Anti-53BP1 (Bethyl, A300-272A). Immunoprecipitated DNA was subjected to real-time qPCR with PerfeCTa SYBR Green SuperMix (Quanta), using a CFX Connect Real-Time Cycler (BioRad). Primers used for the DSB-reporter locus were: for: GGAAGATGTCCCTTGTATCACCAT; rev: TGGTTGTCAACAGAGTAGAAAGTGAA.

Detection of RPA-positive cells[36,37] was performed using a BD FACSCanto 10 flow cytometer. Data were analyzed using FlowJo v10 software (BD). The neutral comet assay was performed per manufacturer's instructions using the Comet Assay Kit (Trevigen, 4250-050) and olive tail moment was analyzed using CometScore 2.0. For the DR-GFP HR assay[24], GFP-positive cells were detected by flow cytometry 3 days after I-SceI transfection.

**DNA fiber assay**. Cells were incubated as indicated with 100 μM CldU and 100 μM IdU. Cells were harvested and DNA fibers were obtained using the FiberPrep kit (Genomic Vision, EXT-001). DNA fibers were stretched on glass coverslips (Genomic Vision, COV-002-RUO) using the FiberComb Molecular Combing instrument (Genomic Vision, MCS-001). Slides were incubated with primary antibodies: Anti-BrdU BU1/75 (Abcam, 6326) for detection of CIdU;

Anti-BrdU B44 (BD, 347580) for detection of IdU; Anti-single-stranded DNA (Millipore Sigma, MAB3034) for detection of DNA. Slides were washed with PBS, and incubated with secondary antibodies: Anti-mouse Cy3.5 preadsorbed (Abcam, 6946); Anti-rat Cy5 preadsorbed (Abcam, 6565); Anti-rabbit BV480-conjugated (BD Biosciences, 564879). Slides were mounted and imaged with a Leica SP5 confocal microscope and analyzed using LASX 3.3.0.16799 software.

**Immunofluorescence**. Cells were washed with PBS three times before fixation with 4% paraformaldehyde for 10 min. After another three washes with PBS, cells were permeabilized using 0.2% Triton X-100 for 10 min. Cells were then washed twice with PBS and blocked for 10 min with 3% BSA in PBS. Next, cells were incubated with primary antibody (53BP1: Bethyl, A300-272A, 1:2000; or γH2AX: Bethyl, A300-081A, 1:250) for 2 h. After three washes with PBS, cells were incubated for 1 h in secondary antibody (AlexaFluor 488: Invitrogen, A11008, 1:2000). All antibodies were diluted to their final concentrations in 3% BSA in PBS and all incubations were performed at room temperature. Slides were mounted with DAPI-containing Vectashield mounting medium (Vector Labs) and imaged using a DeltaVision Elite Deconvolution Inverted microscope. The images were analyzed using ImageJ 1.52p software. The area of damage was defined using mCherry signal, then mean 53BP1 or γH2AX intensity within the area was quantified[27].

**Genomic and transcriptomic analysis**. Genomic and transcriptomic data for ovarian cancer samples were obtained from the Australian Ovarian Cancer Study cohort (OV-AU), which is part of the International Cancer Genome Consortium (ICGC). Somatic structural variants (duplications, deletions, inversions, intra- and inter-chromosomal translocations) were identified using qSV[56] from the whole-genome sequencing data, and length of homology was reported for the structural variation junctions. The structural variations were classified as NHEJ-mediated if there were ≤3 bp homologous sequences. There were 48–2431 (median: 292) structural variants per sample in the ovarian cohort. RNA-sequencing-based expression data were also available for these tumors from the ICGC data portal (https://dcc.icgc.org/). We grouped the samples into four categories based on combinatorial high (above median) and low (below median) expression of BRCA2 and KAT5; this enabled us to analyze samples with BRCA deficiency due to classic BRCA2 inactivation or loss as well as those potentially mediated by other mechanisms. Survival statistical analysis was performed based on the TCGA Ovarian cancer samples[57] with and without BRCA2 mutations using R survival package (version 2.41-3).

**Statistics and reproducibility**. For the neutral comet and immunofluorescence assays, as well as the replication fork degradation DNA fiber experiments, the Mann–Whitney statistical test was performed. For all other assays, the statistical analysis performed was the *t*-test (two-tailed, unequal variance unless indicated). Statistical significance is indicated for each graph (ns = not significant, for $p > 0.05$; * for $p \leq 0.05$; ** for $p \leq 0.01$; *** for $p \leq 0.001$; **** for $p \leq 0.0001$). The exact p-values are listed for each figure in the Source Data file. Statistical analyses were performed using GraphPad Prism 8 or Microsoft Excel 15 software. Western blot experiments were reproduced at least two times.

**Reporting summary**. Further information on research design is available in the Nature Research Reporting Summary linked to this article.

## Data availability
Data supporting the findings of this manuscript are available from the corresponding author upon reasonable request. Read counts for all screens are provided in Supplementary Data files. The source data underlying Figs. 1b, d, f, 2a–c, 3a–f, 4a–d, 5a–d, 6a–c, 7a–c, and Supplementary Figs. 1a, b, 2a, b, 3a, b, 4a, b, 6a–f, 7a, b, 8a, b, 9a–c, 10a, b, 11a, b, 13a, b are provided as a Source Data file. Genomic and transcriptomic data for the OV-AU cohort was obtained from https://dcc.icgc.org/projects/OV-AU and https://xenabrowser.net/datapages/?hub=https://tcga.xenahubs.net:443. Source data are provided with this paper.

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

## Acknowledgements

We would like to thank Drs. Jeremy Stark, Roger Greenberg, Robert Brosh, Mariano Russo, Jacob Hornick, Yunsung Kim, and Michael O'Connor for materials and advice; and the following Penn State College of Medicine core facilities: Flow Cytometry, Genomic Analyses, and Imaging. Experimental design schemes and summary models were created with Biorender.com. This work was supported by: NIH R01ES026184, NIH R01GM134681 and the St. Baldrick's Foundation (to G.L.M.), NIH F31CA243301 (to E.M.S.), and NIH R01GM129066 (to S.D.).

## Author contributions

K.E.C., E.M.S., T.T., A.H., A.D., N.J.T., C.M.N., and G.L.M. conducted the experiments; X.L. and H.G.W. provided the CRISPR library lentiviral preparation; Y.I.K. supported the computational analyses of the CRISPR screen results; A.S. and S.D. performed the computational analyses of the OV-AU and OV-TCGA datasets; K.E.C. and G.L.M. designed the experiments and wrote the paper.

## Competing interests

The authors declare no competing interests.
