## [Peer Review File · Nature Communications]

Editorial Note: Parts of this Peer Review File have been redacted as indicated to remove personal communications where no permission to publish could be obtained

Reviewers' comments:

Reviewer #1 (Remarks to the Author):

In this study by Clements et al, the authors conducted three independent CRISPR screens (two ko and one activation screens) to identify novel targets that contribute to PARP inhibitor (olaparib) sensitivity in parental HeLa cells and isogenic clone with BRCA2 KO. The screen identified some known or expected targets and uncovered several targets that were not previously linked to PARP inhibitor resistance. Specifically, they found that the deletion of HUWE1 or TIP60 significantly attenuated olaparib sensitivity in BRCA2 KO HeLa cells. Focusing then specifically on TIP60, the authors then tested whether the TIP60 loss associated PARP inhibitor resistance depends on any known PARP resistant mechanisms, including release PARP1 from chromatin, reduce apparently increased fork speeds, restoring HR and resection. Specifically, they found that Tip60 loss restored the PARP inhibitor resistance in BRCA2 KO HeLa cells require the presence of 53BP1 and REV7. Further experiments led to a model, in which TIP 60 deletion in BRCA2 deficient cells increases 53BP1 binding around the olaparib induced breaks to suppress excess end- resection and single-strand annealing.

Overall, the study identifies new players in PARP inhibitor resistance in BRCA2 deficient cells. The findings complement the existing data on how BRCA1 deficient cells gain PARP inhibitor resistance. The seemingly opposite role of 53BP1 on PARP inhibitor sensitivity in BRCA1- vs BRCA2- deficient cells provide new insights on the unique role of these two BRCA proteins in DNA repair and homologous recombination. In contrast to the strong genetic observations, the mechanistic model requires a few additional experiments for support. Specifically,

1) Throughout the experiments, the authors only used one PARP inhibitor – olaparib. Given the potential different biological effects of different PARP inhibitors, the Tip60 rescue should be validated with at least another PARP inhibitors.

2) What is the effect of 53BP1 deletion (alone) on the olaparib sensitivity of BRCA2 KO HeLa cells? If the model is true, would 53BP1 deletion further sensitize BRCA2 KO cells to PARP inhibitor? How about REV7 deficiency?

3) Increase 53BP1 binding to DNA breaks 4 hours after break induction (Fig 5A) is a key mechanistic finding in this manuscript. Prior cell biology studies suggest that 53BP1 foci form at 3-10 minutes after laser-induced DSB and gradually decrease after DNA repair (6-20 hours). The increased 53BP1 binding at 4 hours after break-induction could be caused by i) delayed repair or ii) increased recruitment (as the author suggested). Kinetic studies about 53BP1 recruitment together with DNA cutting/repair would be crucial to establish the increased 53BP1 recruitment. If the model is true, the authors might expect increased 53BP1 recruitment even in WT cells with Tip60 knockdown. Is that true? On this note, (optional) RPA-CHIP or RAD51 CHIP surrounding the break might complement the flow analyses to solidify the impacts of TIP60 deletion on DNA end resection.

4) Lastly, in Fig5C, the authors curated DNA repair signature in the Australian ovarian cancer study cohort public database as evidence for increased NHEJ. The NHEJ signature is derived from somatic structural variants accumulated in the lifetime of the tumor and defined as 0-3 homology. Given the importance of increase NHEJ in explaining the reduced olaparib induced DSB in the proposed model, a direct measure for NHEJ, such as using the new NHEJ reporters published on Nature Communication in 2018 by Dr. Stark would be helpful.

Minor Comments:

1) Fig 1, the dose (so-called low dose) of olaparib in panel A is 5uM and the in panel C and E (so-called high dose) is 4uM. I found the "low" vs "high" confusing. It would be cleaner to directly mark the actual dose of olaparib or the IC in there.

2) The chromatin association of PARP1 (Fig S5A) should be improved.

3) TIP60 is an acetyltransferase. Does restore TIP60 expression and activity (WT or catalytically inactive) sufficient to reverse the rescue?

Reviewer #2 (Remarks to the Author):

This is an interesting paper describing TIP60 as a factor whose depletion reverses PARP inhibitor sensitivity in BRCA2 deficient cells.

By performing CRISPR knockout screens and activation screen using HeLa cells, the authors identified TIP60 as a factor whose depletion leads to PARP inhibitor resistance in BRCA2-deficient cells.

Although the finding that TIP60 depletion leads to PARP inhibitor resistance in BRCA2-deficient cells is interesting, the study is incomplete.

Major points:

- 1) Figure 3A, B, C: The authors should perform rescue experiments using siRNA –resistant cDNA of wild type Tip60 and a catalytic dead mutant of Tip60, in order to determine whether acetyltransferase activity of Tip60 is critical for this phenotype.
- 2) Figure 3A, B, C: The authors should use BRCA1-deficient cells as well as BRCA2-deficient cells for these studies.
- 3) Figure 3A, B, C: Similarly, the authors should perform rescue experiments using siRNA – resistant cDNA of wild type HUWE1 and a catalytic dead mutant of HUWE1, in order to determine whether ubiquitin ligase activity of HUWE1 is critical for this phenotype.
- 4) The authors did not follow up HUWE1 for mechanistic studies.
- 5) The authors should test whether replication fork stability in BRCA1/2-deficient cells is restored by depletion of Tip60.
- 6) Figure 4B, C: The authors should use at least two different siRNAs for each gene (TIP60, REV7).
- 7) Figure 5D: How can increased NHEJ lead to PARP inhibitor resistance in BRCA2-deficient cells? The authors did not show that NHEJ is increased in TIP60-depleted BRCA2-deficient cells. The authors did not show that “increased” NHEJ is critical for PARP inhibitor resistance in TIP60-depleted BRCA2-deficient cells.

Reviewer #3 (Remarks to the Author):

Specific comments

Q. Why use HeLa cells (cervical cancer) rather than breast, pancreas or ovary given these are the indications for Olaparib use clinically?

Q. Figure 2B – BRCA2 ko sensitizes HeLa cells to Olaparib – why are neither BRCA1 or BRCA2 identified as significant sensitizing hits in the low dose sensitization screen?

Q. What was the rationale for selecting the 4uM drug concentration and 4-day timepoint? This combination of a high drug dose (IC90) and a short duration of experiment is unusual for a resistance screen where typically experiments are 10-12 doubling times (usually 2-3 weeks) to better capture the range of resistance genes. The risk with picking extremes of drug dose and experiment duration is that many smaller effect size resistance genes will be missed and in particular for pathway enrichment analyses may help identify the important networks that drive resistance.

Q. Why were KAT5 and HUWE1 selected for further investigation – in particular as the former is ranked 123 in the gene list. This is important to discuss as a significant challenge in the analysis of genome-wide CRISPR screens is the systematic analysis of data using prior knowledge e.g. pathway enrichments, druggability, cancer relevance of a gene etc.

Q. Did the authors consider using in their gene set enrichment analysis KEGG or Reactome – these can give additional insights into biological pathways. Additionally, if the significant hits from the CRISPRn and CRISPRa screens are combined and gene set enrichment analyses re-run, are there any additional GO or gene sets that show enrichment ?

Q. What was the rationale for using Stealth siRNA for the validation gene knockdowns versus CRISPR/Cas9 using different sgRNA sequences to those in the Brunello library? This should be included in the manuscript as either approach has drawbacks, and siRNA typically used together with CRISPR ko.

Q. After an initial validation of Olaparib resistance when HUWE1 is knocked out with siRNA, there is no further exploration of the mechanisms behind this and I'd rewrite sentence below to reflect this:

From the PARPi resistance screen in BRCA2KO cells, two DNA repair genes previously unconnected to PARPi resistance, namely the histone acetyltransferase TIP60 (KAT5) and E3 ubiquitin ligase HUWE1, drew our attention.

Q. Q. In the Supp tables of the genes ranked for effect following the RSA analysis – what does the column 'Log(sgRNA Fold Change) of the Best Guide' mean?

Minor corrections

Page 7 - Depletion of TIP60 or HUWE1 rescues PARPi sensitivity in BRCA2-deficient cells – for clarity I would suggest changing rescues to something like 'confers resistance'

Response to referees

We would like to thank the reviewers for their constructive comments. To address the reviewers' concerns, we are submitting a substantially revised manuscript with 16 new figure panels. We made our best effort to address the reviewer's comments in the current COVID-19 pandemic situation, with our laboratory being closed for the past 6 weeks, and no re-opening scheduled for the foreseeable future. In this unprecedented situation, we sincerely hope that the reviewers will find our revised manuscript suitable. Below, please find our point-by-point reply to reviewers' comments (our responses in red font)

Reviewer #1

We are glad that the reviewer found that our manuscript “*identifies new players*” and “*provides new insights*”, through “*strong genetic observations*”. We thank the reviewer for their helpful comments. We have addressed these comments in the revised manuscript as indicated below:

1) Throughout the experiments, the authors only used one PARP inhibitor – olaparib. Given the potential different biological effects of different PARP inhibitors, the Tip60 rescue should be validated with at least another PARP inhibitors.

We now show in the **new Supplementary Fig. 7a, b** that loss of TIP60 also rescues the sensitivity of BRCA2-deficient cells to two additional PARP inhibitors, namely veliparib and niraparib. This indicates that the response is not specific to olaparib, but instead broadly applies to multiple PARP inhibitors.

2) What is the effect of 53BP1 deletion (alone) on the olaparib sensitivity of BRCA2 KO HeLa cells? If the model is true, would 53BP1 deletion further sensitize BRCA2 KO cells to PARP inhibitor? How about REV7 deficiency?

The effects of depletion of 53BP1 alone (Fig. 5c) or REV7 alone (Fig. 5d) were in fact presented in the manuscript. 53BP1 knockdown does not increase the olaparib sensitivity of BRCA2-deficient cells, whereas depletion of REV7 mildly increases it. Based on previous literature (PMID: 23377543, 30297459) and our own findings (Fig. 6), in the presence of TIP60, recruitment of 53BP1 to double strand breaks is already reduced. This potentially explains why 53BP1 knockdown in TIP60-proficient cells does not further increase olaparib sensitivity.

3) Increase 53BP1 binding to DNA breaks 4 hours after break induction (Fig 5A) is a key mechanistic finding in this manuscript. Prior cell biology studies suggest that 53BP1 foci form at 3-10 minutes after laser-induced DSB and gradually decrease after DNA repair (6-20 hours). The increased 53BP1 binding at 4 hours after break-induction could be caused by i) delayed repair or ii) increased recruitment (as the author suggested). Kinetic studies about 53BP1 recruitment together with DNA cutting/repair would be crucial to establish the increased 53BP1 recruitment. If the model is true, the authors might expect increased 53BP1 recruitment even in WT cells with Tip60 knockdown. Is that true? On this note, (optional) RPA-CHIP or RAD51

CHIP surrounding the break might complement the flow analyses to solidify the impacts of TIP60 deletion on DNA end resection.

The experimental system employed here is different from laser-induced DSBs, as it involves enzymatic induction of DSB upon expression of a site-specific nuclease (via 4-OHT and Shield-1L treatment). Importantly, this ensures that the amount of damage is the same among different genetic conditions, so the increase in 53BP1 cannot be explained by an increase in the amount of damage.

The time course of 53BP1 recruitment to enzymatic-induced DSBs is likely to be different than that of direct laser DSB induction.

[REDACTED]

Nevertheless, to address the reviewer comment that the increased 53BP1 binding may be attributed to either increased recruitment or delayed repair, we investigated 53BP1 recruitment by immunofluorescence at earlier time points. At 10 minutes after treatment, we observed very few instances of mCherry foci (which mark nuclease recruitment to chromatin) indicating that this timepoint is too early as no DNA damage occurs in our system. At intermediate timepoints—1 hour and 2.5 hour after treatment—we analyzed the mean intensity of 53BP1 and γ H2AX at these discrete sites of damage indicated by mCherry foci (**new Supplementary Fig. 10a, b**). We observed an increase in 53BP1 intensity at these sites over time regardless of TIP60 status. While there was no difference in 53BP1 intensity between genetic conditions at the 1-hour post-induction timepoint, at 2.5 hours after treatment we observed a trend towards greater 53BP1 accumulation TIP60-depleted cells. This indicates that the increase in 53BP1 binding upon TIP60 depletion that we observed in the ChIP assay at 5-hours after treatment is likely due to a gradual increase in 53BP1 accumulation at the DSB over time.

At both timepoints studied, we found that cells depleted of TIP60 showed an increase in γ H2AX intensity per mCherry focus. This is consistent with previous reports that TIP60 depletion leads to increased retention (rather than increased binding) of γ H2AX at sites of damage (PMID: 20479123, 26438602, 17709392). However, both control and TIP60-depleted cells showed similar 1.25-fold increases in mean γ H2AX intensity from the 1-hr to 2.5-hr timepoints, indicating that the kinetics of repair are similar within the timeframe in which 53BP1 is beginning to accumulate. This argues against a delayed repair being the reason for the increased 53BP1 foci in TIP60-depleted cells. This is also in line with the comet assay results we present in Fig. 5b showing that TIP60 knockdown reduces olaparib-induced DSB formation in BRCA2-knockout cells, indicating that loss of TIP60 does not delay DNA repair.

Finally, we would like to mention that, as the reviewer mentions, the Greenberg lab has indeed previously shown increased 53BP1 recruitment in wildtype (BRCA2-proficient) cells upon TIP60 knockdown (PMID: 23377543, 30297459), consistent with our model.

4) Lastly, in Fig5C, the authors curated DNA repair signature in the Australian ovarian cancer study cohort public database as evidence for increased NHEJ. The NHEJ signature is derived from somatic structural variants accumulated in the lifetime of the tumor and defined as 0-3 homology. Given the importance of increase NHEJ in explaining the reduced olaparib induced DSB in the proposed model, a direct measure for NHEJ, such as using the new NHEJ reporters published on Nature Communication in 2018 by Dr. Stark would be helpful.

Unfortunately, the challenge in using the NHEJ reporter mentioned by the reviewer (EJ7-GFP) developed by Dr. Jeremy Stark's laboratory is that the role of 53BP1 in specific NHEJ outcomes measured by this assay has remained elusive. These assays measure EJ between blunt Cas9-induced DSBs without indel mutations, which the Stark lab has shown to require XLF, XRCC4, and KU70 (PMID: 29950655). However, when they analyzed 53BP1, they found that this factor was dispensable for this NHEJ event, using amplicon sequencing of a rearrangement between two CAS9 DSBs (PMID: 28057860, Figure 4B).

[REDACTED]

Thus, these assays unfortunately cannot be used to evaluate 53BP1-dependent NHEJ.

Minor Comments:

1) Fig 1, the dose (so-called low dose) of olaparib in panel A is 5uM and the in panel C and E (so-called high dose) is 4uM. I found the "low" vs "high" confusing. It would be cleaner to directly mark the actual dose of olaparib or the IC in there.

We now mention the exact dose on the figure, rather than "low" and "high".

2) The chromatin association of PARP1 (Fig S5A) should be improved.

We repeated this experiment and now present a result of improved quality in the **new Fig. 5a**.

3) TIP60 is an acetyltransferase. Does restore TIP60 expression and activity (WT or catalytically inactive) sufficient to reverse the rescue?

As mentioned right below in the response to Reviewer #2 comment 1, our results that 53BP1 is required for olaparib resistance mediated by TIP60 depletion, strongly suggest that the catalytic activity of TIP60 is involved in this mechanism, as 53BP1 inhibition by TIP60 is mediated by the histone acetylation activity of TIP60. Unfortunately, it is technically very challenging to address this experimentally as TIP60 knockout or long-term depletion results in loss of viability. As described below, we provide a figure for reviewers showing the results of preliminary experiments using a TIP60 conditional knockout (cKO) mouse embryonic fibroblast (MEF) cell line expressing either wildtype or catalytically inactive TIP60, showing that the catalytic activity of TIP60 is essential for olaparib sensitivity in BRCA2-deficient cells. Unfortunately, because our laboratory was shut down in the context of the COVID-19 pandemic, we were unable to perform enough replicates and statistical analyses to be able to include this data in the revised manuscript.

Reviewer #2

We are glad that the reviewer found that our submission is "*an interesting paper*". We thank the reviewer for their helpful comments. We have addressed these comments in the revised manuscript as indicated below:

1) Figure 3A, B, C: The authors should perform rescue experiments using siRNA –resistant cDNA of wild type Tip60 and a catalytic dead mutant of Tip60, in order to determine whether acetyltransferase activity of Tip60 is critical for this phenotype.

Our results clearly show that the impact of TIP60 loss on olaparib sensitivity is mediated by 53BP1 and the associated REV7-Shieldin complex (Fig. 5c, d). It has been established in the literature that TIP60 regulates binding of 53BP1 to chromatin by catalyzing H2AK15 and H4K16 acetylation (PMID: 27153538, PMID: 23377543). Thus, it is highly likely that the olaparib resistance mechanism reported here depends on the loss of the catalytic activity of TIP60. We now comment on this in the Discussion section.

We agree with the reviewer that experiments employing the catalytic inactive mutant of TIP60 would be informative. Unfortunately, these experiments are technically very challenging.

[REDACTED]

TIP60 knockout or long-term depletion results in loss of viability. Thus, there is no easily available experimental setup that would allow us to directly test the role of the catalytic activity in human cells.

Nevertheless, in an attempt to address the reviewer’s comment, we obtained from Dr. Roger Greenberg a conditional TIP60-knockout (cKO) mouse embryonic fibroblast (MEF) cell line, expressing either exogenous HA-tagged wildtype (termed “WT”) or catalytic inactive (Q377E/G380E, termed “HD”) human TIP60 cDNA (published in Li et al, PMID: 30297459). Treatment with 4-hydroxytamoxifen (4’OHT) for 3 days results in deletion of the floxed endogenous TIP60 gene, resulting in expression of exogenous WT or HD human TIP60 cDNA as the only source of TIP60 in these cells. Before our laboratory was shut down in the context of the COVID-19 pandemic, we managed to perform preliminary experiments with these cell lines. As shown in the attached figure, in two independent experiments, BRCA2 depletion in HD cells resulted in a much less olaparib sensitivity than BRCA2 depletion in WT cells, indicating that the catalytic activity of TIP60 is required for the olaparib sensitivity of BRCA2-deficient cells, in line with our model. Unfortunately, the lack of enough replicates for statistical analyses precludes the inclusion of this data in the revised manuscript.

2) Figure 3A, B, C: The authors should use BRCA1-deficient cells as well as BRCA2-deficient cells for these studies.

As we describe in the Discussion section, it was previously shown that loss of 53BP1 rescues the olaparib sensitivity of BRCA1-deficient cells (PMID: 25799992, 20362325). In contrast, we show here that loss of 53BP1 does not alter the olaparib sensitivity of BRCA2-deficient cells, while loss of TIP60, an antagonist of 53BP1, rescues it (Fig. 5c, d). Thus, our results highlight a previously unappreciated difference between BRCA1 and BRCA2, likely explained by the different steps of the homologous recombination pathway in which the two proteins participate.

3) Figure 3A, B, C: Similarly, the authors should perform rescue experiments using siRNA – resistant cDNA of wild type HUWE1 and a catalytic dead mutant of HUWE1, in order to determine whether ubiquitin ligase activity of HUWE1 is critical for this phenotype.

HUWE1 is a very long protein, of 4374 aminoacids and thus its cDNA is 13kbp long. Unfortunately, despite our best efforts, performing mutagenesis and the other necessary studies with such a large construct was not feasible within the available timeframe. However, we have performed a number of mechanistic studies to address the reviewers comment #4, as described below.

4) The authors did not follow up HUWE1 for mechanistic studies.

We now significantly expanded the revised manuscript to include a section on HUWE1 mechanistic studies. We now show in the **new Fig. 4a-d, and new Supplementary Fig. 6a,b** that loss of HUWE1 results in increased RAD51 expression, which partially rescues the homologous recombination defect of BRCA2-deficient cells. These findings indicate that HUWE1 and TIP60 have different mechanisms of action, and further broaden the relevance of our study, which now describes not one but two mechanisms of PARP inhibitor resistance in BRCA2-deficient cells.

5) The authors should test whether replication fork stability in BRCA1/2-deficient cells is restored by depletion of Tip60.

We now show that depletion of TIP60 also rescues the fork stability defect of BRCA2-deficient cells (**new Fig. 7b**). As we comment in the revised Discussion section, there is considerable debate in the field regarding the exact contribution of fork protection to olaparib resistance, even though a correlation with chemoresistance has been clearly demonstrated.

6) Figure 4B, C: The authors should use at least two different siRNAs for each gene (Tip60, REV7).

We present two different siRNAs for TIP60 and HUWE1 in Supplementary Fig. 4a-d.

7) Figure 5D: How can increased NHEJ lead to PARP inhibitor resistance in BRCA2-deficient cells? The authors did not show that NHEJ is increased in TIP60-depleted BRCA2-deficient cells. The authors did not show that “increased” NHEJ is critical for PARP inhibitor resistance in TIP60-depleted BRCA2-deficient cells.

Unfortunately, as mentioned in the response to Reviewer #1 comment 4, no experimental assay is available to measure 53BP1-dependent NHEJ, thus we cannot directly address this point. As we mentioned in the figure legend and in the Discussion section, increased NHEJ is a proposed model and we do not rule out that other mechanisms may be involved or may explain the observed findings. We believe that our model is reasonable because:

- 1) We show that 53BP1 is increased in TIP60-depleted cells (Fig. 6a, new Supplementary Fig. 10), and 53BP1 is essential for olaparib resistance upon TIP60 depletion (Fig. 5c). 53BP1 has been clearly shown in the literature to suppress end resection, but simply blocking end resection by CTIP depletion does not restore PARPi resistance (Supplementary Figure 12). Thus, it is plausible that TIP60 inhibits a role of 53BP1 in promoting NHEJ, and this activity is released upon TIP60 depletion.
- 2) In fact, we do show that TIP60-deficient patient tumors show an increase in the non-homologous end joining pathway of double-strand break repair (Fig. 6c), consistent with the expected effects of an increase in 53BP1 binding at the double-strand break.

Reviewer #3

We thank the reviewer for their helpful comments. We have addressed these comments in the revised manuscript as indicated below:

Q. Why use HeLa cells (cervical cancer) rather than breast, pancreas or ovary given these are the indications for Olaparib use clinically?

The main reason we did the screen in HeLa cells is that we wanted to employ a clean genetic system with an isogenic pair of wildtype-BRCA2 and knockout-BRCA2 cell lines. As we explain in the revised manuscript, this setup allows us to directly compare the results from the various screens in the two cell lines, and eliminates potentially confounding issues caused by mutations found in patient-derived BRCA2-mutant cell lines. We originally attempted to create an isogenic BRCA2 wildtype/knockout pair in multiple cell lines, but we only managed to obtain it in HeLa cells. HeLa cells are amenable to transfection and lentiviral transduction, and have a good proliferation rate -thus they are ideal for these kinds of genetic screens. Moreover, basic mechanisms of DNA repair are conserved in HeLa cells, which have been used as a model in the field for a long time. In line with this, we validated the phenotypes and the functional insights for the top screen hits in multiple cell lines, confirming the validity of the screen.

Q. Figure 2B – BRCA2 ko sensitizes HeLa cells to Olaparib – why are neither BRCA1 or BRCA2 identified as significant sensitizing hits in the low dose sensitization screen?

There are several potential reasons why this may occur, including poor guide function and lethality of the knockout (in the case of BRCA1). While BRCA1 and BRCA2 did not appear as top hits, pathway analyses presented in Supplementary Fig. 1 shows that DSB repair (ranked 2nd) and Recombination (ranked 6th) are among the top biological processes in the sensitization screen. Indeed, many BRCA1/2 interactors including RAD51 were among the top hits from this screen.

Q. What was the rationale for selecting the 4 μ M drug concentration and 4-day timepoint? This combination of a high drug dose (IC₉₀) and a short duration of experiment is unusual for a resistance screen where typically experiments are 10-12 doubling times (usually 2-3 weeks) to better capture the range of resistance genes. The risk with picking extremes of drug dose and expt duration is that many smaller effect size resistance genes will be missed and in particular for pathway enrichment analyses may help identify the important networks that drive resistance.

We agree with the reviewer that there are several different ways to perform a drug resistance screen. We cannot rule out that we missed some of the smaller effect size genes using our experimental setup, but this is a risk with any screen setup. As we describe in our manuscript, many of the top hits in the various screens were expected based on previous literature, and we have validated some of the new ones. Thus, we think that our experimental setup is relevant and, even beyond the small number of top hits we focused on, our large-scale data is highly informative.

Q. Why were KAT5 and HUWE1 selected for further investigation – in particular as the former is ranked 123 in the gene list. This is important to discuss as a significant challenge in the analysis of genome-wide CRISPR screens is the systematic analysis of data using prior knowledge e.g. pathway enrichments, druggability, cancer relevance of a gene etc.

As we now explain in the revised manuscript, we decided to investigate these two proteins as they both have significant roles in cellular processes related to DNA replication and repair and are potential interactors of PARP1 (PMID: 22267412, 26976643), yet they have not been directly connected to PARPi sensitivity.

Q. Did the authors consider using in their gene set enrichment analysis KEGG or Reactome – these can give additional insights into biological pathways. Additionally, if the significant hits from the CRISPRn and CRISPRa screens are combined and gene set enrichment analyses re-run, are there any additional GO or gene sets that show enrichment ?

We now include KEGG pathway analyses in the **new Supplementary Fig. 1b and 2b**. We also include a gene set enrichment analysis of the combined knockout and activation resistance screens, performed with both GO and KEGG terms, in the **new Supplementary Fig. 4a, b**. We also performed Reactome analyses but did not include them as they did not reveal any additional insights.

Q. What was the rationale for using Stealth siRNA for the validation gene knockdowns versus CRISPR/Cas9 using different sgRNA sequences to those in the Brunello library? This should be included in the manuscript as either approach has drawbacks, and siRNA typically used together with CRISPR ko.

The siRNA approach allowed us to confirm the screen results in multiple cell lines rapidly and cost-effectively, as making knockout cell lines is time-consuming and not all cell lines may be amenable to gene editing. Moreover, this approach also allowed us to perform mechanistic

experiments in reporter cell lines such as U2OS-DSB and U2OS DR-GFP.

In the revised manuscript, we are including experiments with HUWE1 knockout cells generated by CRISPR using sequences different from those in the library (**new Fig. 4a, and new Supplementary Fig. 6a**). In this setup also, loss of HUWE1 confers olaparib resistance to BRCA2-deficient cells -thus validating our siRNA experiments.

Q. After an initial validation of Olaparib resistance when HUWE1 is knocked out with siRNA, there is no further exploration of the mechanisms behind this and I'd rewrite sentence below to reflect this:

From the PARPi resistance screen in BRCA2KO cells, two DNA repair genes previously unconnected to PARPi resistance, namely the histone acetyltransferase TIP60 (KAT5) and E3 ubiquitin ligase HUWE1, drew our attention.

We now significantly expanded the revised manuscript to include a section on HUWE1 mechanistic studies. We now show in the **new Fig. 4a-d, and new Supplementary Fig. 6a, b** that loss of HUWE1 results in increased RAD51 expression, which partially rescues the homologous recombination defect of BRCA2-deficient cells. These findings indicate that HUWE1 and TIP60 have different mechanisms of action, and further broaden the relevance of our study, which now describes not one but two mechanisms of PARP inhibitor resistance in BRCA2-deficient cells.

Q. Q. In the Supp tables of the genes ranked for effect following the RSA analysis – what does the column ‘Log(sgRNA Fold Change) of the Best Guide’ mean?

We apologize for the lack of clarity. Each gene was targeted by multiple guide RNAs (sgRNAs) within the libraries. The fold change in representation between the olaparib- and DMSO- treated conditions were calculated for each individual sgRNA. The indicated column represents the fold change in abundance of the guide RNA which showed the greatest change from the untreated to the treated condition. In the revised manuscript we changed the column heading to: “*Log(Fold Change in Abundance) for the Best Guide*”.

Minor corrections

Page 7 - Depletion of TIP60 or HUWE1 rescues PARPi sensitivity in BRCA2-deficient cells – for clarity I would suggest changing rescues to something like ‘confers resistance’

As indicated by the reviewer, we changed the sentence to: “*Depletion of TIP60 or HUWE1 confers PARPi resistance in BRCA2-deficient cells.*”

REVIEWER COMMENTS

Reviewer #1 (Remarks to the Author):

Given the COVID19 situation, we think that the authors have addressed most of our major concerns in the conditions allowed.

Reviewer #2 (Remarks to the Author):

The authors have addressed most of the concerns raised by me.

1) The authors have not directly tested the effect of TIP60 depletion on PARP inhibitor sensitivity of BRCA1-deficient cells. I think that the authors should clearly say that they have not directly tested this in the discussion about difference between BRCA1 and BRCA2.

Reviewer #3 (Remarks to the Author):

* Can I request that authors provide the raw read counts for all CRISPR screens either as additional Supp Tables or via a link to a publicly accessible data storage solution, in the interests of FAIR data policies and data reproducibility efforts.

Please see attached word document for comments on the authors' rebuttal.

Reviewer #3

Below in **red** are the authors response to my initial comments – I have added my own reply (**blue**) to each of these responses below (italicised for clarity):

Q. Why use HeLa cells (cervical cancer) rather than breast, pancreas or ovary given these are the indications for Olaparib use clinically?

The main reason we did the screen in HeLa cells is that we wanted to employ a clean genetic system with an isogenic pair of wildtype-BRCA2 and knockout-BRCA2 cell lines. As we explain in the revised manuscript, this setup allows us to directly compare the results from the various screens in the two cell lines and eliminates potentially confounding issues caused by mutations found in patient-derived BRCA2-mutant cell lines. We originally attempted to create an isogenic BRCA2 wildtype/knockout pair in multiple cell lines, but we only managed to obtain it in HeLa cells. HeLa cells are amenable to transfection and lentiviral transduction and have a good proliferation rate -thus they are ideal for these kinds of genetic screens. Moreover, basic mechanisms of DNA repair are conserved in HeLa cells, which have been used as a model in the field for a long time. In line with this, we validated the phenotypes and the functional insights for the top screen hits in multiple cell lines, confirming the validity of the screen.

Given the large number of BRCA2 wild-type breast, ovarian and pancreatic cell lines, I don't accept the rationale above for using HeLa cells. Any of these cell lines would equally have provided a clean isogenic background but in the right tissue context. CRISPR has been used and published in several BRCA2 wild-type cell lines to create isogenic lines.

Q. Figure 2B – BRCA2 ko sensitizes HeLa cells to Olaparib – why are neither BRCA1 or BRCA2 identified as significant sensitizing hits in the low dose sensitization screen?

There are several potential reasons why this may occur, including poor guide function and lethality of the knockout (in the case of BRCA1). While BRCA1 and BRCA2 did not appear as top hits, pathway analyses presented in Supplementary Fig. 1 shows that DSB repair (ranked 2nd) and Recombination (ranked 6th) are among the top biological processes in the sensitization screen. Indeed, many BRCA1/2 interactors including RAD51 were among the top hits from this screen.

This goes to the main issue I have with the experimental design and explains this finding above – a 4-day duration experiment (esp for sensitizing and resistance genes in DDR pathways) is too short – Fig 2a shows clearly that the BRCA2 ko is not lethal in the HeLa model but is sensitizing, and yet it fails to appear as a hit in the CRISPR sensitizing screen (Fig 1a) – while the authors are correct that they validate the hits in subsequent sections of the paper, similar to BRCA2, they are likely missing other important resistance genes – the 4-day data certainly provides interesting data notwithstanding but this feels like an incomplete exploration of the resistance landscape.

Q. What was the rationale for selecting the 4uM drug concentration and 4-day timepoint? This combination of a high drug dose (IC90) and a short duration of experiment is unusual for a resistance screen where typically experiments are 10-12 doubling times (usually 2-3 weeks) to better capture the range of resistance genes. The risk with picking extremes of drug dose and expt duration is that many smaller effect size resistance genes will be missed and in particular for pathway enrichment analyses may help identify the important networks that drive resistance.

We agree with the reviewer that there are several different ways to perform a drug resistance screen. We cannot rule out that we missed some of the smaller effect size genes using our experimental setup, but this is a risk with any screen setup. As we describe in our

manuscript, many of the top hits in the various screens were expected based on previous literature, and we have validated some of the new ones. Thus, we think that our experimental setup is relevant and, even beyond the small number of top hits we focused on, our large-scale data is highly informative.

See comment above.

Q. Why were KAT5 and HUWE1 selected for further investigation – in particular as the former is ranked 123 in the gene list. This is important to discuss as a significant challenge in the analysis of genome-wide CRISPR screens is the systematic analysis of data using prior knowledge e.g. pathway enrichments, druggability, cancer relevance of a gene etc.

As we now explain in the revised manuscript, we decided to investigate these two proteins as they both have significant roles in cellular processes related to DNA replication and repair and are potential interactors of PARP1 (PMID: 22267412, 26976643), yet they have not been directly connected to PARPi sensitivity.

My question has not been answered here sufficiently – what was the process whereby an enriched gene ranked 123rd was selected for further study? Did the authors manually PubMed every single gene until they reached KAT5? It's not clear from the paper what the process was and it's also clear from Supp Fig. 2 that processes such as DNA replication and repair did not appear in either the GO or KEGG gene set enrichment analyses. In the absence of a more logical reason for the selection of KAT5, such studies give the impression that the gene was selected based on some other prior knowledge that has not been included in the paper.

Q. Did the authors consider using in their gene set enrichment analysis KEGG or Reactome – these can give additional insights into biological pathways. Additionally, if the significant hits from the CRISPRn and CRISPRa screens are combined and gene set enrichment analyses rerun, are there any additional GO or gene sets that show enrichment ?

We now include KEGG pathway analyses in the **new Supplementary Fig. 1b and 2b**. We also include a gene set enrichment analysis of the combined knockout and activation resistance screens, performed with both GO and KEGG terms, in the **new Supplementary Fig. 4a, b**. We also performed Reactome analyses but did not include them as they did not reveal any additional insights.

See comment above regarding gene set enrichment analyses.

Q. What was the rationale for using Stealth siRNA for the validation gene knockdowns versus CRISPR/Cas9 using different sgRNA sequences to those in the Brunello library? This should be included in the manuscript as either approach has drawbacks, and siRNA typically used together with CRISPR ko.

The siRNA approach allowed us to confirm the screen results in multiple cell lines rapidly and cost-effectively, as making knockout cell lines is time-consuming and not all cell lines may be amenable to gene editing. Moreover, this approach also allowed us to perform mechanistic experiments in reporter cell lines such as U2OS-DSB and U2OS DR-GFP. In the revised manuscript, we are including experiments with HUWE1 knockout cells generated by CRISPR using sequences different from those in the library (**new Fig. 4a, and new Supplementary Fig. 6a**). In this setup also, loss of HUWE1 confers olaparib resistance to BRCA2-deficient cells -thus validating our siRNA experiments.

Thank you.

Q. After an initial validation of Olaparib resistance when HUWE1 is knocked out with siRNA,

there is no further exploration of the mechanisms behind this and I'd rewrite sentence below to reflect this: From the PARPi resistance screen in BRCA2KO cells, two DNA repair genes previously unconnected to PARPi resistance, namely the histone acetyltransferase TIP60 (KAT5) and E3 ubiquitin ligase HUWE1, drew our attention.

We now significantly expanded the revised manuscript to include a section on HUWE1 mechanistic studies. We now show in the **new Fig. 4a-d, and new Supplementary Fig. 6a, b** that loss of HUWE1 results in increased RAD51 expression, which partially rescues the homologous recombination defect of BRCA2-deficient cells. These findings indicate that HUWE1 and TIP60 have different mechanisms of action, and further broaden the relevance of our study, which now describes not one but two mechanisms of PARP inhibitor resistance in BRCA2-deficient cells.

Thank you.

Q. In the Supp tables of the genes ranked for effect following the RSA analysis – what does the column 'Log(sgRNA Fold Change) of the Best Guide' mean?

We apologize for the lack of clarity. Each gene was targeted by multiple guide RNAs (sgRNAs) within the libraries. The fold change in representation between the olaparib- and DMSO- treated conditions were calculated for each individual sgRNA. The indicated column represents the fold change in abundance of the guide RNA which showed the greatest change from the untreated to the treated condition. In the revised manuscript we changed the column heading to: "Log(Fold Change in Abundance) for the Best Guide".

If RSA was used in these analyses, then standard outputs such as median values for control vs treatment samples, RSA rank and Activity Rank would be expected in the Supp Tables for the three CRISPR screens. I have a major issue with presenting the 'Log(sgRNA Fold Change) of the Best Guide' in these tables – esp if the summary statistics were generated from the values of all the gRNA effects. We include multiple sgRNA per gene in part to mitigate the risk of off-target activity of any one sgRNA. I suspect the best sgRNA fold change has been used because the mean/median of all sgRNA is less impressive and likely due to the short 4-day duration of the experiment. This is not best practice for presenting the output of analyses of CRISPR datasets.

Minor typos

In legend for Fig. 3., "All graphs reflect the average of three experiments performed after 72 hours of treatment...". This I assume does not cover 3f (clonogenic survival assay) which should be indicated as a 2-week experiment.

Response to referees

We would like to thank the reviewers for their positive comments. Below, we address the remaining comments of reviewers 2 and 3 (our responses in red font).

Reviewer #2

1) The authors have not directly tested the effect of TIP60 depletion on PARP inhibitor sensitivity of BRCA1-deficient cells. I think that the authors should clearly say that they have not directly tested this in the discussion about difference between BRCA1 and BRCA2.

We agree with the reviewer and we now include this statement as indicated.

Reviewer #3

Can I request that authors provide the raw read counts for all CRISPR screens either as additional Supp Tables or via a link to a publicly accessible data storage solution, in the interests of FAIR data policies and data reproducibility efforts.

In the new Supplementary Tables 2 and 3, we provide the raw read counts for each of the guides for all screens, as well as the individual guides' activities and RSA ranks.

Please see attached word document for comments on the authors' rebuttal.

Given the large number of BRCA2 wild-type breast, ovarian and pancreatic cell lines, I don't accept the rationale above for using HeLa cells. Any of these cell lines would equally have provided a clean isogenic background but in the right tissue context. CRISPR has been used and published in several BRCA2 wild-type cell lines to create isogenic lines.

We agree with the reviewer that performing the screen in HeLa cells is a potential limitation of our study. In the Discussion section of the revised manuscript, we now point out this limitation, and state that hits specific to breast, ovarian or pancreatic cancer may have been missed because of this.

This goes to the main issue I have with the experimental design and explains this finding above – a 4-day duration experiment (esp for sensitizing and resistance genes in DDR pathways) is too short – Fig 2a shows clearly that the BRCA2 ko is not lethal in the HeLa model but is sensitizing, and yet it fails to appear as a hit in the CRISPR sensitizing screen (Fig 1a) – while the authors are correct that they validate the hits in subsequent sections of the paper, similar to BRCA2, they are likely missing other important resistance genes – the 4-day data certainly provides interesting data notwithstanding but this feels like an incomplete exploration of the resistance landscape.

We agree with the reviewer that the experimental setup is another limitation of our study, and state this in the Discussion section of the revised manuscript. We also indicate that additional resistance genes may have been missed because of this.

My question has not been answered here sufficiently – what was the process whereby an enriched gene ranked 123rd was selected for further study? Did the authors manually PubMed every single gene until they reached KAT5? It's not clear from the paper what the process was and it's also clear from Supp Fig. 2 that processes such as DNA replication and repair did not appear in either the GO or KEGG gene set enrichment analyses. In the absence of a more logical reason for the selection of KAT5, such studies give the impression that the gene was selected based on some other prior knowledge that has not been included in the paper.

We did not have a prior knowledge or other reasoning not included in the paper for selecting TIP60 for validation. In fact, our approach was more in line with what the reviewer mentioned: we manually searched the literature for all top200 hits in order to select those with relevant DNA repair functions. We now describe this in the revised manuscript.

If RSA was used in these analyses, then standard outputs such as median values for control vs treatment samples, RSA rank and Activity Rank would be expected in the Supp Tables for the three CRISPR screens. I have a major issue with presenting the 'Log(sgRNA Fold Change) of the Best Guide' in these tables – esp if the summary statistics were generated from the values of all the gRNA effects. We include multiple sgRNA per gene in part to mitigate the risk of off-target activity of any one sgRNA. I suspect the best sgRNA fold change has been used because the mean/median of all sgRNA is less impressive and likely due to the short 4-day duration of the experiment. This is not best practice for presenting the output of analyses of CRISPR datasets.

In the new Supplementary Tables 2 and 3, we provide both the raw read counts for each of the guides for all screens, as well as the original outputs from each RSA analysis, including the activity and RSA rank of every guide.

In legend for Fig. 3., "All graphs reflect the average of three experiments performed after 72 hours of treatment...". This I assume does not cover 3f (clonogenic survival assay) which should be indicated as a 2-week experiment.

As we described in the Methods section, the clonogenic experiments were performed by incubating the cells with the drug for 72 hours, followed by media change and incubation for 2 weeks. Thus, the figure legend is accurate.

REVIEWER COMMENTS

Reviewer #1 (Remarks to the Author):

Overall, the screens and the initial validations of the screen hits, including Tip60 and the HUWE1, are convincing. The results are informative for the field and provides independent validation and comparison on different cell types.

I do have some concerns with the specific mechanisms proposed for HUWE1 and Tip60. For HUWE1, the authors show Rad 51 protein levels increases and they quoted their own paper to support Rad51 overexpression can restore HR. However, whether Rad51 increase is the mechanism by which HUWE1 siRNA rescued PARPi sensitivity was not directly tested. For example, it is not clear how HUWE1 increases RAD51 mRNA, by preventing degradation, increase transcription, or indirectly affect cell cycle (RAD51 accumulates in S G2 phase and low in G1 arrested cells) and whether HUWE1's effect RAD51 and HR is specific to BRCA2 deficient cells or true for all cells. In the case of Tip60, the authors show that the rescue by Tip60 siRNA can be partially reversed by 53BP1 siRNA. But this "correlative" data does not necessarily indicate a "cause-effective" relationship. The author cited reference 20, which has made a similar "correlative" connection between Tip60 and chromatin accumulation of PARP1. PARP1 recognizes and binds single-strand DNA nicks and DNA double-strand breaks. There is evidence that PAPP1 replace H1 during transcription. Again, a specific mechanism is not detailed or tested. In general, I felt the screen is informative to the field and the selected targets seem robust. The authors made good efforts to probe the next step. However, direct genetic tests (such as mutations, rescues) or biochemical analyses, would be nice to provide firm mechanistic insights. On the other hand, I do understand the difficulty in conducting any experiments during institution shutdown or even during the slow ramping up process. So if the authors to soft their claim and acknowledge the correlative nature of the results, it would be fine to me. The screen is self remains valuable to the field.

Reviewer #3 (Remarks to the Author):

Thank you for your replies.

It is good to see that the authors have submitted the raw read counts for the CRISPR screens in the interests of reproducibility - it has however exposed another technical weakness which I had not appreciated - that none of the screens were run as replicates. The industry standard for most CRISPR screens (even in technically challenging T-cells would be biological duplicates - Shifrut, Cell 2018). In the absence of replicates, there will be more false positives. This is especially pertinent given one of my main concerns with the paper (below) - that in a genome-wide screen for modifiers of response to PARP inhibitors the gene selected for further investigation was ranked 123rd for resistance - the authors response (that they searched the literature for the top 200 for any connection to DNA repair) strains credulity. It's pertinent that gene set enrichment analyses using KEGG and GO did not reveal enrichment for DDR pathways.

A cherry picking hit gene approach to the analysis of CRISPR screens would have been acceptable a few years ago when we were all still in a steep learning curve as to the successful technical execution of these. Nowadays, one would hope for a more sophisticated approach and more robust experimental design. That being said, if my DDR expert colleagues believe that discovery of KAT5 (TIP60) is an important discovery in this space then I am happy to bow to their expertise and support (tepidly) publication.

My question has not been answered here sufficiently – what was the process whereby an enriched gene ranked 123rd was selected for further study? Did the authors manually PubMed every single gene until they reached KAT5? It's not clear from the paper what the process was and it's also clear from Supp Fig. 2 that processes such as DNA replication and repair did not appear in

either the GO or KEGG gene set enrichment analyses. In the absence of a more logical reason for the selection of KAT5, such studies give the impression that the gene was selected based on some other prior knowledge that has not been included in the paper.

Finally, the HUGO nomenclature for TIP60 is KAT5 (and it is even listed as KAT5 in the CRISPR gene lists the authors provide) - I assume it would be preferable to try and adhere to this as a naming convention throughout the paper?

Response to referees

We would like to thank the reviewers for their comments. Below, we address the reviewer comments (our responses in red font).

Reviewer #1

In general, I felt the screen is informative to the field and the selected targets seem robust. The authors made good efforts to probe the next step. However, direct genetic tests (such as mutations, rescues) or biochemical analyses, would be nice to provide firm mechanistic insights. On the other hand, I do understand the difficulty in conducting any experiments during institution shutdown or even during the slow ramping up process. So if the authors to soft their claim and acknowledge the correlative nature of the results, it would be fine to me. The screen is self remains valuable to the field.

We are very happy to learn that the reviewer found our screens to be valuable to the field, and our manuscript to be acceptable for publication. We thank the reviewer for their comments. In the revised manuscript, we incorporated the reviewer's feedback and present a more careful interpretation of our results.

Reviewer #3

It is good to see that the authors have submitted the raw read counts for the CRISPR screens in the interests of reproducibility - it has however exposed another technical weakness which I had not appreciated - that none of the screens were run as replicates. The industry standard for most CRISPR screens (even in technically challenging T-cells would be biological duplicates - Shifrut, Cell 2018). In the absence of replicates, there will be more false positives. This is especially pertinent given one of my main concerns with the paper (below) - that in a genome-wide screen for modifiers of response to PARP inhibitors the gene selected for further investigation was ranked 123rd for resistance - the authors response (that they searched the literature for the top 200 for any connection to DNA repair) strains credulity. It's pertinent that gene set enrichment analyses using KEGG and GO did not reveal enrichment for DDR pathways.

A cherry picking hit gene approach to the analysis of CRISPR screens would have been acceptable a few years ago when we were all still in a steep learning curve as to the successful technical execution of these. Nowadays, one would hope for a more sophisticated approach and more robust experimental design. That being said, if my DDR expert colleagues believe that discovery of KAT5 (TIP60) is an important discovery in this space then I am happy to bow to their expertise and support (tepidly) publication.

My question has not been answered here sufficiently – what was the process whereby an enriched gene ranked 123rd was selected for further study? Did the authors manually PubMed every single gene until they reached KAT5? It's not clear from the paper what the process was and it's also clear from Supp Fig. 2 that processes such as DNA replication and repair did not appear in either the GO or KEGG gene set enrichment analyses. In the absence of a more logical reason for the selection of KAT5, such studies give the impression that the gene was selected based on some other prior knowledge that has not been included in the paper.

We thank the reviewer for their thoughtful comments and appreciate their input, which, as described below, resulted in a significant improvement of our manuscript by incorporating a newly performed replicate for each of the screens in our manuscript.

First, we would like to mention that our original knockout screens were indeed performed a few years ago, in 2017, and thus they were using the conceptual design and data analysis tools available at that time. Since our lab's expertise is in DNA replication and damage repair studies, we decided to embark on these screens anticipating that DNA replication and repair pathways may be enriched among the top hits, and that we would be able to focus on this area for follow-up analyses, while revealing additional new pathways that others with different expertise might pursue after our publication. We were surprised that RNA and mitochondria processes featured much more prominently among the top hits, but nevertheless, because of our expertise, we went along with our original plan to identify DNA replication and repair proteins among the top hits. TIP60/KAT5 and HUWE1 were among a few of those who caught our eye, because they have important DNA repair roles however they have not been associated with PARPi resistance so far, so we selected them for subsequent validation. Since we could successfully validate them with siRNA-knockdown, this indicated to us that they represented true hits, and thus we proceeded with designing follow-up mechanistic analyses. We are certainly not trying to deceive the reviewers or the readers. It may appear somewhat inelegant in retrospect given the subsequent developments within the field, but this was truly our approach at that time.

Nevertheless, we agree with the reviewer that it has become standard in the CRISPR screening field to perform biological replicates of screens. Thus, for this third revision of our manuscript, we performed an independent replicate of each of the three screens in our manuscript. Moreover, we have now bioinformatically analyzed the screens results with MAGeCK software rather than RSA, which was used in our initial submission. Unlike RSA, which was initially developed for RNAi-based screens, MAGeCK was developed specifically for CRISPR screens. Moreover, MAGeCK allows the combined analysis of replicate screens, generating a single gene ranking list with statistical analyses. We present in the new Supplementary Tables 1-3 the read counts for each screen replicate, and the MAGeCK analysis output listing scores, p-values and additional parameters for each guide and gene. The read counts presented in these above-mentioned tables indicate a good accord between the two replicates of each screen.

Importantly, for all three screens, performing a second replicate and evaluating both replicates by combined MAGeCK analyses resulted in a much better representation of DNA replication and repair processes among the top hits (new Supplementary Figures 1-4). For the CRISPR knockout screen for olaparib sensitivity in wildtype cells, upon the combined analysis, all top biological processes are now DNA damage-related, as other processes identified based on the original screen only and presented in our previous submission (RNA processing, translation etc) have dropped out (new Supplementary Figure 1). In the case of the knockout screen for olaparib resistance in BRCA2^{KO} cells, DNA replication and repair processes feature prominently within the top pathways, while not being present in the original single screen analysis in our previous submission (new Supplementary Figure 2). These results are much more in line with our original expectations, and we believe highlight the significance of our screens for the field. Thus, performing the screen replicates and analyzing them by combined MAGeCK analysis greatly improved our manuscript, and we are thankful to the reviewer for pointing out that performing a single replicate was a weakness of our previous submission.

Importantly, HUWE1 and TIP60/KAT5 continue to feature prominently as significant hits in the combined MAGeCK analysis of the replicate knockout screens for olaparib resistance in

BRCA2^{KO} cells. As explained in our revised manuscript, HUWE1 and TIP60/KAT5 are among the 214 genes which were most highly significantly enriched in this screen (FDR value lower than 5%). This validates our original selection of these two genes for validation and follow-up mechanistic experiments. Moreover, we found that, out of these top 214 hits, 23 genes are associated with DNA replication and repair processes (new Supplementary Figure 5). These include, besides HUWE1 and TIP60/KAT5, a number of genes previously associated with olaparib resistance (e.g. E2F7, PARP1) as well as a number of other genes not previously connected to this. This further highlights the relevance of our work to the field.

We hope that the reviewer appreciates that we have made tremendous efforts to perform the replicate screens and the combined analyses for this third revision, under the current COVID-19 laboratory operating conditions. The results validated our original hit selection and at the same time significantly improved our manuscript, as described above. We sincerely hope that the reviewer will find this third revision of our manuscript to be acceptable for publication.

Finally, the HUGO nomenclature for TIP60 is KAT5 (and it is even listed as KAT5 in the CRISPR gene lists the authors provide) - I assume it would be preferable to try and adhere to this as a naming convention throughout the paper?

We had initially chosen to refer to this gene as TIP60, since this name is more common within the DNA repair field. But we agree with the reviewer that it would be best to follow the HUGO nomenclature which is also present in the CRISPR library files. In the revised version, we have now used the gene name KAT5 instead of TIP60 throughout the manuscript text and figures.

REVIEWERS' COMMENTS

Reviewer #3 (Remarks to the Author):

The authors are to be congratulated for adding replicates to the CRISPR screens and changing analysis. The paper has been considerably improved by this.

Response to referees

Reviewer #3 (Remarks to the Author):

The authors are to be congratulated for adding replicates to the CRISPR screens and changing analysis. The paper has been considerably improved by this.

We are glad that the reviewer found our revision to be satisfactory, and thank the reviewer for their kind words. We moreover thank the reviewer once again for their comments to the previous versions of our manuscript, which significantly improved the paper.